# A mechanistic theory for aquatic food chain length

Colette L. Ward [1,2,3] & Kevin S. McCann[1]

Multiple hypotheses propose an ostensibly disparate array of drivers of food chain length (FCL), with contradictory support from natural settings. Here we posit that the magnitude of vertical energy flux in food webs underlies several drivers of FCL. We show that rising energy flux fuels top-heavy biomass pyramids, promoting omnivory, thereby reducing FCL. We link this theory to commonly evaluated hypotheses for environmental drivers of FCL (productivity, ecosystem size) and demonstrate that effects of these drivers should be context-dependent. We evaluate support for this theory in lake and marine ecosystems and demonstrate that ecosystem size is the most important driver of FCL in low-productivity ecosystems (positive relationship) while productivity is most important in large and high-productivity ecosystems (negative relationship). This work stands in contrast to classical hypotheses, which predict a positive effect of productivity on FCL, and may help reconcile the contradictory nature of published results for drivers of FCL.

[1] Department of Integrative Biology, University of Guelph, 50 Stone Road East, Guelph, ON N1G 2W1, Canada. [2] National Center for Ecological Analysis and Synthesis, University of California, Santa Barbara, 735 State Street, Suite 300, Santa Barbara, CA 93101-5504, USA. [3] Department of Evolutionary Biology and Environmental Studies, University of Zürich, Winterthurerstrasse 190, 8057 Zürich, Switzerland. Correspondence and requests for materials should be addressed to C.L.W. (email: ward@nceas.ucsb.edu)

Ecologists have long sought to understand the nature and origin of variation in food chain length (FCL; the maximum trophic position among all members of a food web) due to its implications for trophic control, nutrient cycling, and bioaccumulation of environmental contaminants. Multiple hypotheses propose a diverse array of potential drivers of FCL, including resource availability[1–5], the dynamic stability of food web configurations[6], body size and physiological design constraints[7], body size scaling[8], ecosystem type[9,10], intraguild predation[11–14], ecosystem size[15–18], and productive space (total ecosystem productivity adjusted for ecosystem size)[10].

Most hypotheses for FCL are supported by meta-analysis[19] and experimental evidence from simple in vitro and in silico systems; however, tests of environmental drivers in natural settings yield contradictory results. FCL is often positively related to ecosystem size[19–24], however, several studies have documented no relationship[25] or a threshold effect[26,27]. Similarly, FCL has been positively related to resource availability[22,25,28,29], although other authors have found no relationship[20,23,30]. Several mechanisms may underlie these results and their relative importance remains unclear. Moreover, drivers of FCL may operate simultaneously or interactively[14] or their relative importance may be context-dependent[31], and there exists little theoretical framework for understanding hypotheses' interconnectedness.

Early theory for FCL assumed that communities are structured as relatively simple linear food chains of dietary specialists and, by extension, that vertical change in species richness (additions of top predators to trophic chains) is the main mechanism underlying variation in FCL (e.g., refs. [5,32,33]). Post and Takimoto[13] later added that species insertions to trophic chains can also elongate FCL. This mechanism (henceforth the Classical Species Richness mechanism; CSRM) underlies, either wholly or in part, hypotheses related to environmental drivers of FCL. The resource availability hypothesis assumes that energy transfers between trophic levels are inherently inefficient and therefore limiting to

the persistence of higher-order consumers. Rising energy inputs to basal trophic levels, or factors improving the energetic efficiency of consumers at intermediate trophic levels, should therefore result in the addition of successive top trophic levels[2–5,7]. Any positive relationship between productivity and FCL should be driven by species richness unless, among low-productivity ecosystems, rising productivity renders it more beneficial for a predator to feed at intermediate instead of lower trophic levels. The ecosystem size hypothesis is also consistent with the CSRM, although the effect of size may be attributable to other mechanisms. Larger ecosystems often harbor greater species richness, increasing the occupancy likelihood of a novel top predator or intermediate trophic-level consumer capable of elongating food chains[10,15,16].

Since development of the CSRM, theoretical and empirical work has shown that simple linear food chains are not the dominant structures underlying community organization[34,35] and, moreover, are not accurate models for community response to changing environmental conditions[34]. Instead, omnivory and trophic complexity are prevalent[36–38], food web structure is flexible[39–43] and community response to environmental change can occur in the absence of changes in vertical species richness (e.g., refs. [25,27,44]). Such internal change in food web topology and energy flow provides an alternative mechanism by which FCL can vary. Omnivory, in particular, can be important in determining FCL[11,13,14] and can provide a mechanistic link between FCL and environmental conditions[11,17,18,27].

Here we argue that understanding how and under what conditions omnivory responds to environmental gradients may help us understand context-dependency in drivers of FCL and, by extension, contradictory results from tests of FCL drivers in natural settings. We first show that the degree of omnivory among predators is determined by the magnitude of vertical energy transfers within food webs. We develop a simple, general energy-related theory for FCL for the case where species richness

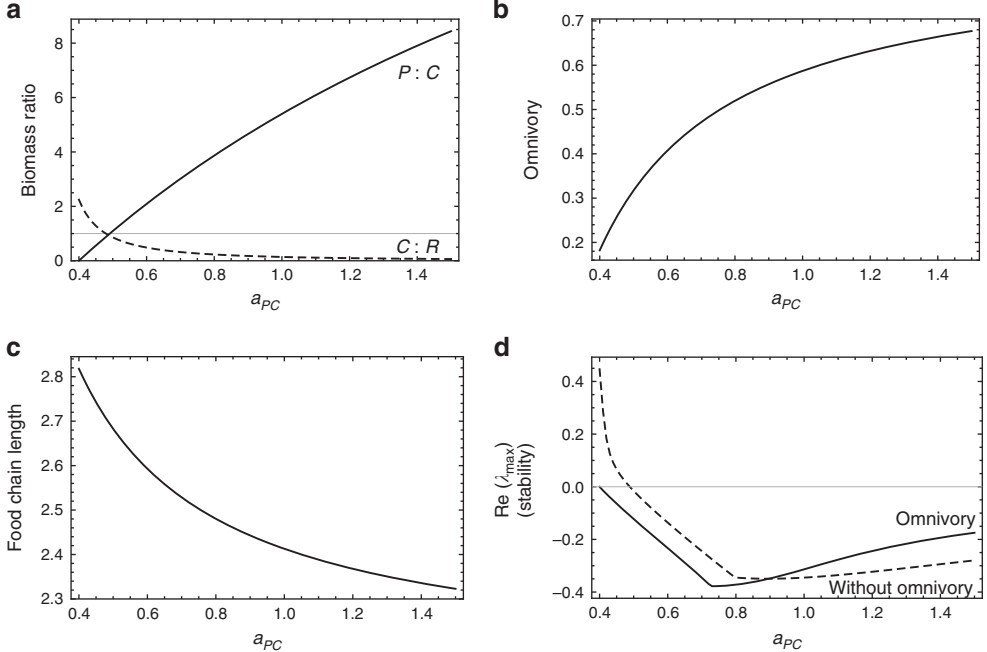

**Fig. 1** Predicting food-chain length from the energy flux mechanism. Equilibrium solutions from Eq. 1 for the effect of increasing $a_{PC}$ on **a** biomass pyramid shape (i.e., biomass ratios of $P$:$C$, and $C$:$R$), **b** the degree to which $P$ is omnivorous, **c** food-chain length, and **d** stability (for systems with and without omnivory), for the following parameters: $r = 2.0$, $K = 3.0$, $a_{CR} = 1.1$, $a_{PR} = 0.2$, $e_{CR} = 1.0$, $e_{PC} = 1.0$, $e_{PR} = 1.0$, $m_C = 0.7$, $m_P = 0.7$. Increasing $e$ or $K$ produces qualitatively similar results

is constant. We use this theory to demonstrate the implications of changing vertical energy flux within food webs (e.g., changing attack rates, resource carrying capacity) on omnivory and FCL. We then demonstrate the utility of this theory by illustrating its congruence with two empirically documented drivers of FCL (productivity and ecosystem size) and show that the relative importance of these drivers should be context-dependent. Finally, we test the predictions of our Energy Flux theory using empirical data from lake and marine food webs. Although this theory is germane across ecosystem types, its predictions are more likely to be realized in aquatic ecosystems, where strong body size scaling in pelagic food chains[45] and the predominance of ectothermic consumers with low metabolic costs (and thus high-trophic conversion efficiency[46]) are likely to promote strong vertical energy flux[47]. It is also most likely to be realized in ecosystems with low-habitat heterogeneity, where prey refugia do not restrict vertical energy flux.

## Results

**An energy flux mechanism for food-chain length**. Here we consider variation in the strength of top-down control that predators exert on consumers, and its implications for biomass pyramid shape and omnivory. We begin with a simple Lotka–Volterra food chain model with the addition of omnivory:

$$dR/dt = rR(1 - R/K) - a_{CR}CR - a_{PR}PR$$
$$dC/dt = e_{CR}a_{CR}CR - m_CC - a_{PC}PC \qquad , \qquad (1)$$
$$dP/dt = e_{PC}a_{PC}PC + e_{PR}a_{PR}PR - m_PP$$

where $r$ is the growth rate of the resource ($R$), $K$ is the carrying capacity of the resource, $a_{CR}$ is the maximum consumption rate of the consumer ($C$) on the resource, $a_{PC}$ and $a_{PR}$ are the maximum consumption rates of the predator ($P$) on the consumer and resource, respectively, $e_{CR}$, $e_{PC}$, and $e_{PR}$ are the conversion efficiencies of consumed biomass into new consumers and predators, and $m_C$ and $m_P$ are the consumer and predator mortality rates. If $P$ consumes its prey as a linear function of prey density, then the degree of $P$'s omnivory (the ratio of $P$'s consumption of $R$ relative to $C$) depends on the equilibrium sizes of $R$ and $C$ (i.e., the relative availability of $R$ and $C$ to $P$).

The magnitude of vertical energy transfers (hereafter 'energy flux') through a food chain can be altered in 2 general ways: (i) changes in the resource's carrying capacity ($K$), and (ii) changes in any of the parameters governing the rate of energy transfer between any consumer and resource pair (i.e., in Eq. 1, attack rate ($a$) or conversion efficiency ($e$)). Regardless of which approach is employed, the theory that follows remains qualitatively the same. This simple realization allows us to generalize our FCL theory to create predictions for any environmental or biological attribute (e.g., ecosystem size and productivity) which can influence energy flux.

Following the approach of Rip and McCann[48], in Fig. 1 we demonstrate that increasing energy flux through a food chain (here, increasing $a_{PC}$ in Eq. 1; increasing $e$ or $K$ yields similar results) gives rise to an increasingly top-heavy biomass pyramid (Fig. 1a) as increasing energy flow inflates $P$, which suppresses $C$ and ultimately allows $R$ to increase. When we allow the food chain to respond to this changing biomass pyramid shape via omnivory, the system effectively 'adapts' as $P$, which we have assumed consumes its prey as a linear function of prey density, increases the relative amount of $R$ consumed relative to $C$ (Fig. 1b), thereby reducing FCL (Fig. 1c). In other words, increasing energy flux causes omnivory to arise passively among predators, shortening FCL[17]. For weak to moderate amounts of omnivory, this omnivorous response tends to be a stabilizing

response to top-heavy biomass pyramid configurations (Fig. 1d), in concurrence with previous findings[49,50]. The strength of this omnivorous response will increase if the top predator is allowed to behave (i.e., if $P$ increases its preference for the more abundant resource $R$); such density dependent behavior, or switching, would exaggerate effects on FCL.

These results are robust to variation in food web structure and functional response form (Supplementary Note 1). Theory will generally yield this answer as long as any process drives biomass accumulation in the top predator, which, in turn, will tend to produce cascading top-down impacts that generate the conditions for increasing omnivory. As such, the result is very general. Moreover, as we explain below, this result is generally robust to additions of weak to moderate density dependence in $C$ and $P$ (bottom-up forcing), and will persist as long as density dependence is not sufficiently strong to prevent biomass build up (top-heaviness) in the biomass pyramid.

**The relationship to existing hypotheses for FCL**. This general theory is linked to hypothesized environmental drivers of FCL. Within ecosystems, reductions in ecosystem size below the foraging scale of top predators can increase the top-down pressure of predators on consumers[17,18]. We make the assumption here that this argument can be extended to the between-ecosystem effect of changing ecosystem size, and thus assume that the effect of declining ecosystem size on FCL can be captured in our framework by increasing $a_{PC}$, the attack rate of predators on consumers. Working from first principles, the theory in refs. [17,18] make mathematical arguments that the more mobile a consumer is that feeds in multiple habitat types (e.g., littoral versus pelagic habitats in aquatic ecosystems), the more its average attack rate increases as these spatially distinct macro-habitats become smaller or closer to each other. Effectively, and ultimately, a mobile consumer 'views' such a mixed habitat as well-mixed at some reduced spatial scale, thus increasing its average consumption rate relative to a larger, more complex habitat arrangement. Although theoretical in its origin, recent empirical evidence at the between-ecosystem scale supports this argument—Tunney et al.[27], working in a between-ecosystem context, found that lake ecosystem food webs appear to become more top-heavy with decreasing lake size, as such an argument would predict. Predators should also exert stronger top-down control on consumers with rising productivity (the Intra-guild Predation hypothesis[11]) and the effect of rising ecosystem productivity can be represented by increasing a surrogate for productivity ($K$, the carrying capacity of the basal resource). Although declining resource palatability with increasing productivity may counter the effect of increasing $K$, declines in palatability would have to largely outweigh increases in productivity for the outcome to be muted. Consequently, our framework predicts that when species richness is constant, declining ecosystem size and rising productivity should both beget increasing energy flux and, by extension, increasingly top-heavy biomass pyramids. As such the R:C ratio is increased, which results in greater omnivory in $P$ and declining FCL.

**Context dependence in environmental drivers of FCL**. We argue that the Energy Flux theory for FCL may help reconcile contradictory results for tests of the resource availability and ecosystem size hypotheses. For simplicity we use a consumer-resource framework to interpret the effects of ecosystem size (attack rate, $a_{PC}$) and productivity (using the surrogate $K$) on FCL. We consider the 2-dimensional case of a predator ($P$) feeding on a consumer ($C$; Fig. 2a). The extension to the 3-dimensional version with $P$, $C$, and $R$ is simple (with either a

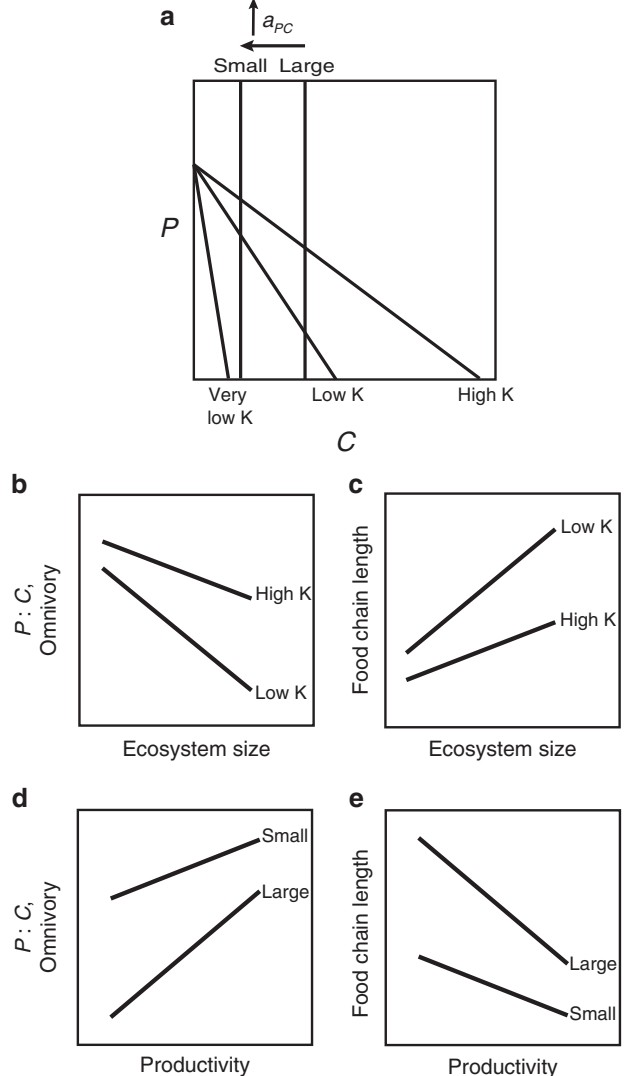

**Fig. 2** Predicting context-dependency in environmental drivers of food-chain length. The simultaneous effects of increasing ecosystem productivity (i.e., increasing $K$) and decreasing ecosystem size (i.e., increasing $a_{PC}$) on biomass pyramid shape, omnivory, and food-chain length. **a** Isoclines of a 2-level Lotka–Volterra model demonstrate equilibrium predator and consumer biomass under simultaneous changes in ecosystem productivity and size. At very low productivity only the consumer is present. **b**, **c** show the effect of ecosystem size, at various levels of productivity, on the degree of top-heaviness (i.e., the ratio of equilibrium predator and consumer biomass), resultant omnivory, and consequent food chain length. **d**, **e** show the same for the effect of productivity at various levels of ecosystem size. As demonstrated in Fig. 1b, top-heavy food webs (i.e., with elevated predator: consumer biomass ratios) promote greater omnivory. **b** demonstrates that productivity should have little effect on food chain length in small systems, but a large effect in large systems. **d** demonstrates that ecosystem size should have a large effect on food chain length in low-productivity systems, but little effect in high-productivity systems

Lotka–Voltrra or Rosenzweig–MacArthur model form) in that whenever the 2-dimensional framework predicts a high $P{:}C$ ratio (i.e., suppressed $C$), $R$ is released (Fig. 1a) in other words, elevated $P{:}C$ ratios imply low $C{:}R$ ratios. Futhermore, as per section (ii) above, lower $C{:}R$ ratios (i.e., Eltonian pyramids) give rise to greater omnivory by $P$.

Our energy flux theory predicts context-dependency in FCL response to simultaneous environmental gradients. Figure 2b, c

show the results of increasing ecosystem size (i.e., decreasing attack rate, $a_{PC}$) on biomass pyramid shape (i.e., $P{:}C$ biomass ratio in Fig. 2b) and FCL (Fig. 2c). For all levels of $K$, increasing ecosystem size reduces the biomass ratio of $P{:}C$ (i.e., renders biomass pyramids more Eltonian; Fig. 2b). This reduction in $P{:}C$ effectively cascades to increase the $C{:}R$ ratio (Fig. 1). Thus, all else equal, increases in ecosystem size render $C$ more abundant relative to $R$ and reduce omnivory, in turn causing FCL to rise with ecosystem size (Fig. 2c). Notably, FCL in high-productivity ecosystems is less influenced by ecosystem size as these ecosystems are already relatively top-heavy and omnivorous. Thus the strength of the effect of changing ecosystem size depends on productivity level—low-productivity ecosystems are most dramatically impacted by increasing ecosystem size (i.e., lower $a_{PC}$). We find a similar result when productivity ($K$) is varied (Fig. 2d, e). Small ecosystems (which already have top-heavy biomass pyramids due to high attack rates) are less influenced by productivity than large ecosystems (Fig. 2d). Therefore changing productivity has a greater impact on FCL in large than small ecosystems (Fig. 2e).

Collectively these results suggest that where species richness is constant: (i) among low-productivity ecosystems, ecosystem size is the dominant driver of FCL, whereas (ii) among high-productivity systems there should be little effect of ecosystem size and productivity should instead be the stronger driver. Additionally, (iii) among small ecosystems there should be little effect of productivity; instead ecosystem size should be the stronger driver of FCL, and (iv) among large ecosystems, productivity should be the dominant driver of FCL and there should be little effect of ecosystem size (Fig. 2b–e). These predictions are robust to additions of bottom-up forcing (e.g., the addition of density dependence to predators). Doing so causes the predator isoclines in Fig. 2a to bend to the right, and the relative shift in $P{:}C$ ratios with changing $K$ and $a_{PC}$ is conserved at all but very high levels of bottom-up forcing.

**A wider view of context dependence in mechanisms driving FCL.** Taken with the CSR mechanism for FCL, these predictions argue for a nonlinear, context-dependent theory for FCL across environmental gradients in aquatic ecosystems (Fig. 3). In effectively 'large' ecosystems (i.e., where $a_{PC}$ is low), FCL should initially rise with increasing productivity due to the CSRM, as depicted in Fig. 3a for very low productivity (via sequential additions of novel predators or insertions of consumers at intermediate trophic levels[5,16,31]). Eventually, the Energy Flux mechanism can begin to play a role when no additional predators or consumers are available or able to colonize the system. This threshold may also be driven by physiological design constraints (sensu ref. [7]), which place upper limits on predator body size and speed, or by the benefits of dietary generalism; both prevent the existence of long chains of specialized predators. At productivity levels above this threshold, rising energy flux causes food webs to become top heavy, which results in passive increases in omnivory among top predators and, consequently, declining FCL. At very high levels of productivity FCL may be mediated once again by species richness or other diversity-related mechanisms, (i) as omnivory becomes strong (i.e., strong intraguild predation, whereby predators eventually extirpate IG-prey), (ii) as environmental conditions related to high productivity (e.g., anoxia) cause local extirpations, or (iii) as species turnover (e.g., rising dominance of inedible producers or consumers[51]) renders increases in energy unavailable for transfer to higher-order consumers. These qualitative predictions concur with those of Post[31], who suggested that that productivity should limit FCL only at very low levels and that environmental drivers of FCL may be context-dependent.

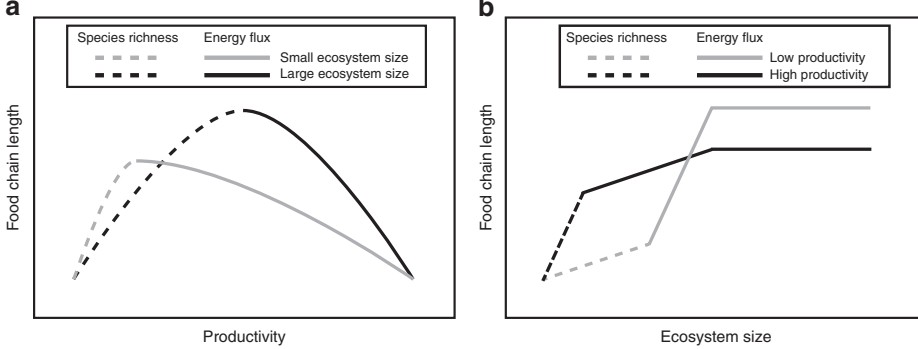

**Fig. 3** Context-dependency in mechanistic drivers of food-chain length **a** in systems with low $a_{PC}$ (effectively 'large' ecosystems) and high $a_{PC}$ (effectively 'small' ecosystems), and **b** in systems with low and high productivity. Dashed lines indicate where food chain length should be set by the Classical Species Richness mechanism; solid lines indicate where food chain length should be set by the Energy Flux mechanism

They also concur with predictions of Post and Takimoto[13], who suggested that FCL may first increase and then decline with increasing resource availability, pursuant to the predictions of Intraguild Predation theory.

In effectively small ecosystems (i.e., ecosystems with high $a_{PC}$), FCL should again initially rise with increasing productivity due to the CSRM (Fig. 3a). However, owing to stronger top-down interactions (attributable to stronger $a_{PC}$ (Fig. 2a), stronger habitat coupling by predators[17,27], or lower diversity[52], predators are able to invade small ecosystems at lower levels of productivity and the threshold at which the Energy Flux mechanism begins to influence FCL occurs at a lower level of productivity than in large ecosystems – this is observed in Fig. 2a by noting that the consumer isocline intersects the predator isocline for small ecosystems at low productivity ($K$), and that productivity ($K$) must be increased in order for the consumer and predator isoclines to intersect for large ecosystems. Additionally, maximum FCL is lower in small ecosystems owing to greater $a_{PC}$ (pursuant to Fig. 2) and should also be lower if smaller ecosystems have lower species richness. At productivity levels greater than this threshold, FCL declines more slowly in small than large ecosystems. This occurs because less change in FCL is possible in small than large ecosystems, because small ecosystems are inherently more top-heavy and have inherently shorter FCL than large ecosystems.

Across a gradient of ecosystem size, FCL should initially increase with ecosystem size owing to the Classical Species Richness mechanism, reaching maximum species richness (where no additional predators or consumers are available or able to colonize the ecosystem) more rapidly in high than low-productivity ecosystems (Fig. 3b). The Energy Flux mechanism thus begins to influence FCL in high-productivity ecosystems at smaller ecosystem size than it does in low-productivity ecosystems. FCL continues to increase with ecosystem size as spatial compression is relieved (i.e., as strong $a_{PC}$ is weakened); pursuant to Fig. 2, FCL changes more rapidly in low-productivity ecosystems. FCL eventually stops changing at very large ecosystem size where spatial compression is entirely relieved.

**Empirical support for the Energy Flux mechanism.** We evaluated support for the Energy Flux mechanism for FCL (predictions in Fig. 1) using a published database of food web data for bounded marine ecosystems[53]. Among these food webs ecosystem volume ranged from $10^{-2}$ to $10^4$ km³, representing an extensive gradient of ecosystem size. Total primary production ranged from 300–9100 t WW km⁻² yr⁻¹ (~30–910 gC m⁻² yr⁻¹), with most food webs having high productivity 1000–9100 t WW

km⁻² yr⁻¹ (~100–910 gC m⁻² yr⁻¹). Under these higher-productivity conditions (here we distinguish between low and high productivity where there occurs an order of magnitude difference in primary production) our theory predicts that top-heaviness and omnivory should be positively related to productivity, and unrelated or weakly related to ecosystem size (Fig. 2). By extension, FCL should be negatively related to productivity and unrelated or weakly related to ecosystem size (Fig. 2b–e). Biomass pyramids became more top-heavy with rising productivity (Fig. 4a) – the log ratio of P:C rose ($p = 0.018$, $R^2 = 0.572$, $F$-value = 9.343, $n = 9$) while the log ratio of C:R declined ($p = 0.077$, $R^2 = 0.34$, $F$-value = 4.128, $n = 10$) across the productivity gradient. Across the ecosystem size gradient (Fig. 4b), only the log ratio of C:R was significantly related to ecosystem size ($p = 0.032$, $R^2 = 0.458$, $F$-value = 6.747, $n = 10$); the log ratio of P:C biomass showed no significant change ($p = 0.376$, $n = 9$). There was a positive and significant relationship between fish omnivory and total primary production ($p = 0.004$, $R^2 = 0.827$, $F$-value = 23.97, $n = 7$; Fig. 4c). Omnivory was not significantly related to ecosystem size ($p = 0.75$, $n = 7$; Fig. 4d). There was no relationship between FCL and total primary productivity, except in ecosystems with high productivity ($p = 0.011$, $R^2 = 0.76$, $F$-value = 15.67, $n = 7$; Fig. 4e). FCL was positively related to ecosystem size ($p = 0.013$, $R^2 = 0.61$, $F$-value = 10.79, $n = 9$; Fig. 4f). We speculate that the result in Fig. 4e (increasing FCL from low to intermediate values of primary production, and decreasing FCL from intermediate to high values) may arise due to a shift from low to high-productivity conditions. As we note above, results for higher-productivity systems (those with primary production >1000 t WW km⁻² yr⁻¹) matched predictions of our Energy Flux theory. Although only two food webs had lower productivity (primary production <400 t WW km⁻² yr⁻¹), we note that results for these systems in Fig. 4a, c, e are consistent with predictions of the Classical Species Richness mechanism, which we suggest is dominant over ranges of low productivity (left side of Fig. 3a). Over this range of productivity one would expect to observe increasing ratios of C:R and P:C (as we observe in Fig. 4a) reflecting species additions or insertions, and omnivory remaining low or increasing to very low levels (as we observe in Fig. 4c). However, we note that we have only nine food webs in total for this analysis, rendering it difficult to determine patterns. Because productivity and ecosystem size were marginally negatively correlated in high-productivity ecosystems for which FCL estimates were available ($p = 0.066$), FCL cannot be ascribed to a unique driver. However, among these high-productivity ecosystems, Akaike's Information Criterion indicated greater support for the relationship between FCL and productivity than that with ecosystem size ($\Delta$AICc = 3.99). Regardless of which environmental

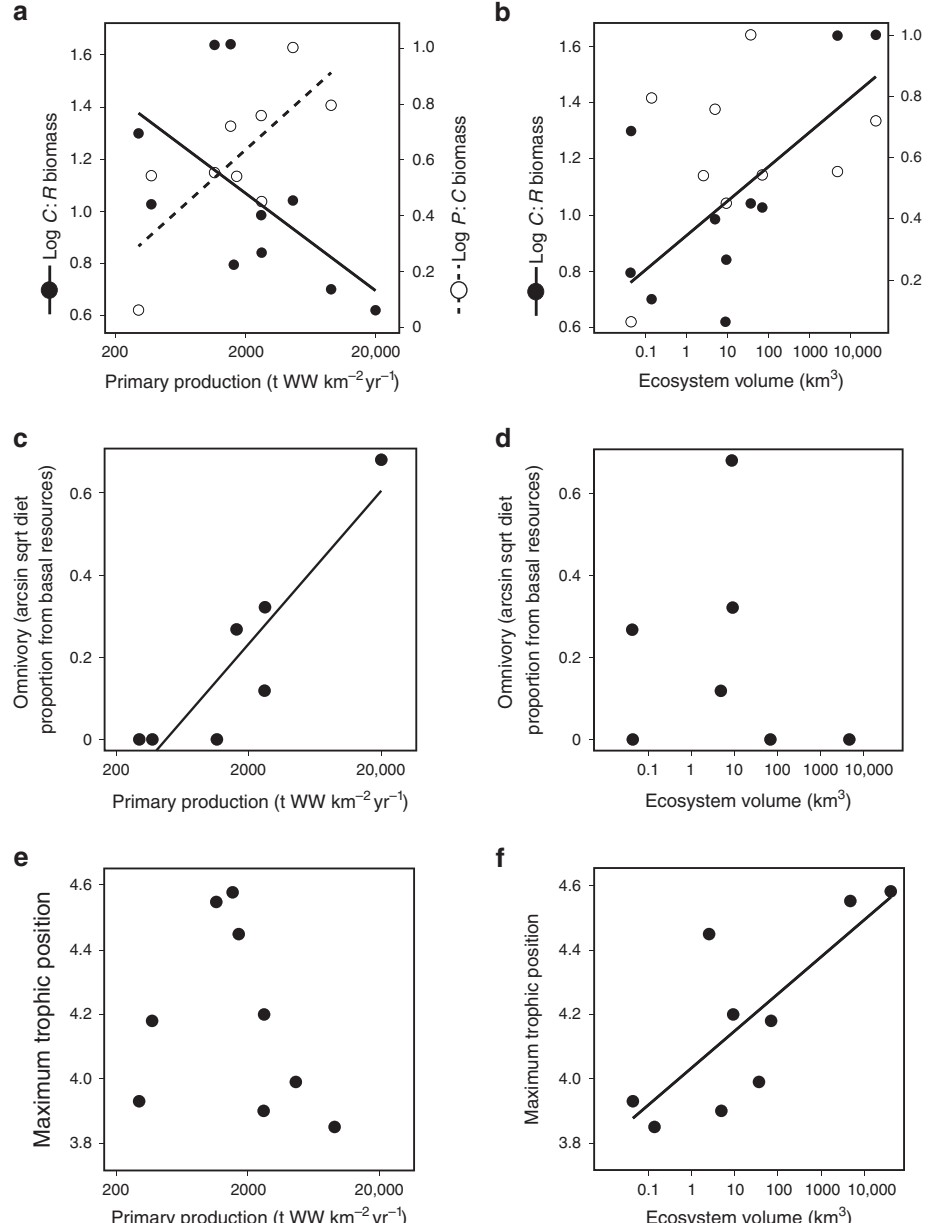

**Fig. 4** Evaluating support for the Energy Flux Mechanism for FCL (predictions in Fig. 1) in marine bounded ecosystems. Relationships between the log ratios of *C:R* biomass (solid lines) and *P:C* biomass (dashed lines) and **a** total primary production and **b** ecosystem size. The relationship between omnivory among fish 35–40 cm and **c** total primary production and **d** ecosystem size. The relationship between food chain length and **e** total primary production, and **f** ecosystem volume for marine bounded systems

driver was most important, the overall result for marine bounded ecosystems is that biomass pyramid shape, omnivory, and FCL were driven by the magnitude of vertical energy flux.

We evaluated predictions arising from Energy Flux theory for context-dependent effects of ecosystem size and productivity on FCL (Fig. 2c, e) using a published database of FCL in lakes[54]. Our results matched most predictions. In oligotrophic and meso-trophic lakes FCL was positively related to ecosystem size ($p \lll$ 0.001, $R^2 = 0.469$, $F$-value = 33.52, $n = 40$; Fig. 5a) and was not related to productivity ($p = 0.28$, $n = 40$; Fig. 5b). In eutrophic lakes FCL was not related to ecosystem size ($p = 0.47$, $n = 23$; Fig. 5a) and was negatively related to productivity ($p = 0.004$, $R^2$ = 0.326, $F$-value = 10.16, $n = 23$; Fig. 5b). Among small lakes FCL was not related to productivity ($p = 0.376$, $n = 33$; Fig. 5c), however, in contrast to predictions, FCL was not related to ecosystem size either ($p = 0.17$, $n = 33$; Fig. 5d). Among large lakes

FCL was negatively related to productivity ($p < 0.001$, $R^2 = 0.348$, $F$-value = 14.93, $n = 30$; Fig. 5c) and showed no relationship with ecosystem size ($p = 0.303$, $n = 30$; Fig. 5d).

## Discussion

A longstanding paradigm underlying our conception of ecosystem response to environmental change invokes changing species richness as the dominant mediator of ecosystem response. However, food web topology is increasingly recognized as an important mediator of ecosystem change, at least over some ranges of environmental gradients[17,27,41], and food web flexibility and adaptive trophic behavior are known to enhance community stability[40,41,55–57]. Omnivory provides one such adaptive and stabilizing mechanism[49,50,57,58], by muting strong, unstable interactions (e.g., high $a_{PC}$) and strengthening others

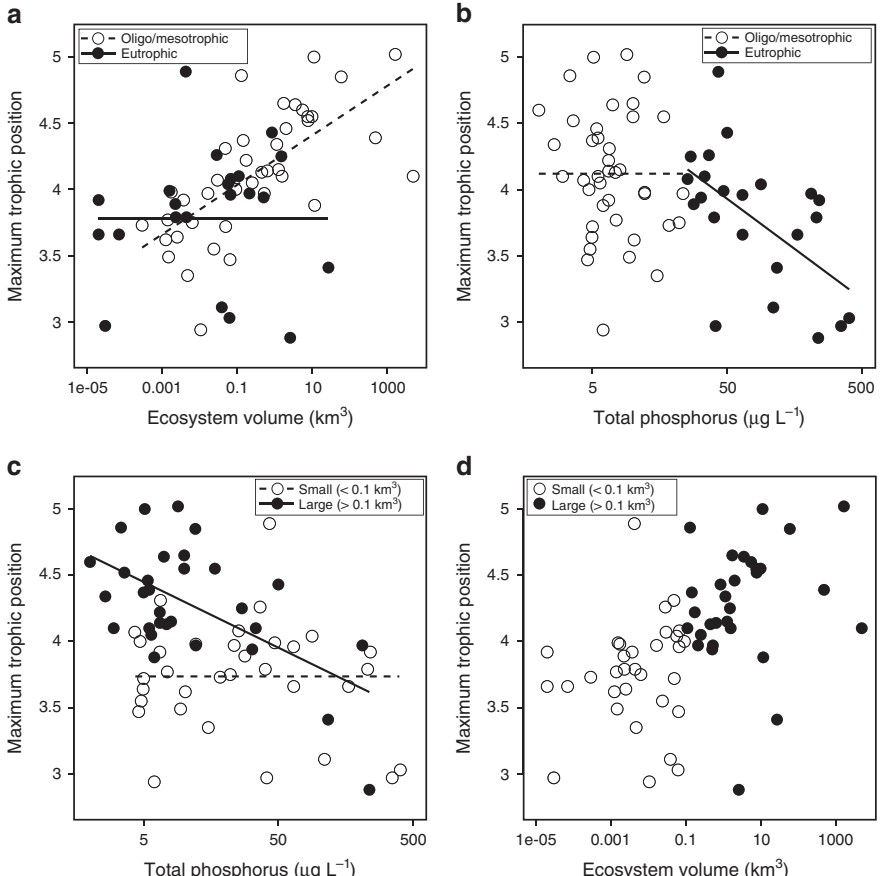

**Fig. 5** Evaluating support for context-dependent effects of ecosystem size and productivity on FCL (predictions in Fig. 2c, e) in lake ecosystems. **a** Food chain length is related to ecosystem volume in oligo- and mesotrophic lakes (dashed line), but not in eutrophic lakes (solid line shown for demonstrative purposes). **b** Food chain length is related to total phosphorus in eutrophic (solid line) but not oligo- and mesotrophic lakes (dashed line shown for demonstrative purposes). **c** Food chain length is related to total phosphorus in large (solid line) but not small lakes (dashed line shown for demonstrative purposes). **d** Food chain length is not related to ecosystem volume in either small or large lakes

(e.g., $a_{PR}$[57]), allowing ecosystem change and adaptation to occur without species loss or community collapse. The widespread prevalence of omnivory in natural settings[38] suggests that this mechanism may be important at a global scale.

Here we document significant effects of food web topology on aquatic FCL across gradients of environmental change and show support for the Energy Flux mechanism. We further show that this mechanism predicts context-dependency in environmental drivers of FCL: ecosystem size should drive FCL in small ecosystems where spatial compression is strongest, and in low to intermediate productivity ecosystems where effects of productivity are weakest. Conversely, productivity should drive FCL in large ecosystems, where the constraints of spatial compression are alleviated, and in highly productive ecosystems.

This context-dependency may explain why many publications have reported different outcomes for tests of drivers of FCL. That FCL is set by a suite of context-dependent drivers was first proposed by Post[31], who suggested that contingencies in successional history and environmental conditions can modify food web structure and attendant FCL from expectations derived from simple linear food chain theory. The preponderance of published evidence supporting a role of ecosystem size, but not productivity, may be due to the frequent use of relatively small study ecosystems (e.g., lakes and islands), where our theory predicts that ecosystem size should be more important. The lack of consistent results for effects of productivity may be attributed to variation in the range and level of productivity employed by various studies. A species richness-based

positive relationship between productivity and FCL should manifest only at very low levels of productivity, where productivity limits colonization by predators (Figs. 2a, 3, and as previously suggested by Post[31]). Theory and empirical results suggest that communities will behave more like simple linear food chains—a requisite condition for the CSRM for FCL to be important–at low productivity[11,51,53]. It is notable that empirical evidence for a positive relationship between productivity and FCL attributed to the CSRM derives largely from simple experimental systems with low species richness (e.g., refs. [59,60]) and from natural settings with low or limiting productivity (e.g., geologically young and/or oligotrophic lakes[28,61], arctic tundra[5,62]). Although we evaluated very few marine ecosystems, it is notable that (i) our marine results (Fig. 4e) concur qualitatively with these expectations and (ii) that the threshold below which there may be a positive relationship between productivity and FCL in our marine ecosystems concurs with the upper limit of the range (100 gC m$^{-2}$ yr$^{-1}$, equivalent to 1000 t WW km$^{-2}$ yr$^{-1}$) proposed by Post[31]. A negative relationship between productivity and FCL has not been documented elsewhere in the literature, likely because the range over which we observed a negative relationship (>1000 t WW km$^{-2}$ yr$^{-1}$ in marine ecosystems and >24 µg Total Phosphorus per L in lakes) has rarely been evaluated, and only then in combination with lower ranges of productivity.

In the absence of species richness estimates at the whole-community level, we are unable to evaluate support for the CSRM (comprising species addition and insertion mechanisms) for FCL.

However, an assessment using only fish species richness (representing intermediate and upper trophic-level consumers) suggests that the species addition component of the CSRM is not supported among our lake data, although we note caveats to this conclusion (Supplementary Note 2). In light of the arguments noted above it is perhaps not surprising that we did not find a relationship between fish species richness and FCL, because most lakes we evaluated were likely not in early successional states and/or limited by productivity.

Our approach adds to a literature documenting mechanisms by which omnivory mediates food web structure and FCL. Building on work suggesting that omnivory (in the context of Intraguild Predation) can determine FCL[13] across gradients of resource availability[11,12], and following a call-to-arms to recognize that drivers of FCL may be context-dependent[31], we show that our energy flux theory allows us to predict the simultaneous and context-dependent effects of ecosystem size and resource availability on FCL. Working in an explicitly metacommunity context, Takimoto et al.[14] made predictions for the simultaneous effects on FCL of colonization/extinction dynamics and the strength of Intraguild Predation across separate gradients of basal productivity, patch density (as a metric of ecosystem size), and disturbance. These authors showed that when Intraguild Predation is strong (in our framework, when biomass pyramids are top-heavy), FCL is shorter than when Intraguild Predation is weak (when biomass pyramids are Eltonian). Here we argue that in an explicitly local context, several hypothesized drivers of food-chain length can act analogously to the omnivory component of IGP (the strength of which is varied by environmental gradients), rather than in tandem with (and in a manner distinct from) Intraguild Predation.

Within food webs consumers may, in contrast to the Energy Flux mechanism described here, become more omnivorous in response to favorable availability of resources at lower trophic levels. This 'bottom-up-driven omnivory' provides an alternative mechanism whereby omnivory-induced changes in FCL can arise in the absence of the vertical Energy Flux mechanism. This phenomenon may be more apparent seasonally in response to the often strongly temporal nature of cross-ecosystem resource subsidies and autochthonous resource availability[63]. It may also be more apparent in particular food web compartments—for example, for the case of parallel pelagic and detritus-based energy channels, increasing primary production may be associated with greater detrital mass[64], resulting in shorter detritus-based food chains.

Our theory also speaks to the longstanding Dynamic Instability hypothesis[6], which suggests that long food chains do not exist because they are dynamically unstable due to the diminishing stabilizing influence of bottom-up controlled basal resources on top predators when food chains are long. Although Sterner et al.[65] documented flaws in the original logic underlying this theory, the Dynamic Instability hypothesis has nevertheless remained a central organizing idea in the FCL literature and is invoked as a mechanism underlying FCL in streams subject to disturbance (e.g., refs. [23,26]). Our results suggest an alternative mechanistic explanation for this hypothesis: that long chains, which might otherwise result from diversity-driven mechanisms under conditions of high vertical energy flux (i.e., high-energy availability in productive and small ecosystems), do not exist because omnivory arises as a passive and stabilizing response to top-heavy biomass pyramids. As ecosystems move along environmental gradients which promote top-heaviness, omnivory first exerts a stabilizing effect on unstable top-heavy biomass pyramids, followed by collapse at extreme ends of these gradients when the predator extirpates the consumer (a manifestation of the Paradox of Enrichment[66], the Dynamic Instability hypotheses[6], and Intraguild Predation[11]). Working in streams, McHugh et al.[26] reported

greater omnivory and declining FCL with increasing disturbance, providing evidence supporting this phenomenon. An additional mechanism that may contribute to the dynamic instability of long food chains, independent of Energy Flux theory, is that rarity may render predators more prone to local extinction in disturbed ecosystems.

We note several caveats to the work presented here. First, as discussed above, we have assumed that the between-ecosystem effect of changing ecosystem size can be represented by a framework for the within-ecosystem effect of declining ecosystem size proposed by refs. [17,18]. An additional caveat is that we use data from ecosystems located in different geographic regions with different regional species pools—each, ostensibly, with a unique suite of relationships between FCL and environmental drivers owing to differences in species identity and regional environmental conditions. We assume that the overall direction and gross shape of relationships between ecosystem-level emergent properties and environmental drivers will be conserved at the between-ecosystem level, albeit with greater variance—for instance, nonlinearities may arise at different environmental thresholds and relationship strengths and elevations may differ owing to between-species physiological, behavioral, and ecological differences in responses to environmental change. This increased variance would, presumably, render it more difficult to detect the relationships we predict. As such, it is notable that our predictions for the Energy Flux theory and the attendant context-dependent nature of FCL are realized in nearly all of our empirical tests in marine and lake ecosystems (Figs. 4, 5) in spite of this variation. For the case where we failed to find the expected relationship between FCL and ecosystem size in small ecosystems (Fig. 5d), we speculate that this greater variance at the between-region level may have influenced this result.

We consider the Energy Flux mechanism—and its implications for context-dependency in environmental drivers within an explicitly local context. This treatment will be sufficient for ecosystems where ecological interactions and material cycling are largely constrained to the local scale. However, the spatial landscape structure in which local communities are inherently embedded can interact with local environmental drivers and influence local ecosystem dynamics and, by extension, FCL in several ways. (i) Theory suggests that colonization/extinction processes arising from metacommunity patch dynamics can influence predictions for FCL[14,16,67]. (ii) In ecosystems permitting the passage of highly mobile predators foraging over very large areas (e.g., migratory species in marine ecosystems), predator biomass, diet, and foraging behavior may not reflect local environmental conditions. (iii) Many ecosystems traditionally regarded as 'closed' routinely receive resource subsidies across ecosystem boundaries (e.g., refs. [68–70]). Collectively these points highlight a largely intractable issue with defining ecosystem boundaries and, by extension, accurately quantifying ecosystem size and resource availability. Incorporating this question into our Energy Flux framework is beyond the scope of this work, and we point the reader toward recent work in the metaecosystem literature, which is beginning to address these ideas[71,72].

In conclusion, we develop theory to show that FCL should arise from an energy flux mechanism in aquatic ecosystems, the context-dependent nature of which should be readily predicted by simple consumer-resource theory. Using data from lake and marine ecosystems, we find that these predictions are largely realized in natural settings: we demonstrate that rising productivity drives declining FCL in large and high-productivity ecosystems (in contrast to predictions of classical hypotheses) and that ecosystem size determines FCL in low-productivity ecosystems, although we failed to observe that ecosystem size drives FCL in small ecosystems. As such, our theory and results may help to reconcile a large literature

of seemingly inconsistent results from aquatic ecosystems. Notably, two primary agents of anthropogenic change–habitat fragmentation and eutrophication are pushing ecosystems toward small and highly productive ecosystems, the very settings where omnivory and eventually 'omnivorous collapse' are predicted to have the greatest impact on FCL.

## Methods

**Food web data**. FCL data for lake ecosystems (Supplementary Data 1) were derived from the database of Vander Zanden and Fetzer[54], who used $^{15}N$ stable isotope data to calculate maximum trophic position among all species present in a food web: $FCL = \frac{\delta^{15}N_{Top\ Predator} - \delta^{15}N_{Baseline\ Consumer}}{3.4} + TP_{Baseline\ Consumer}$. We retained the authors' use of $\Delta^{15}N = 3.4‰$ because there is little evidence for a large effect of varying fractionation factor on FCL over the range of FCL (~3–5) present in our lake data set[73]. We used the database with several modifications: (i) we omitted food webs when there was evidence that a top predator was present >6 months of the year but not included in the food web, (ii) where we found evidence that the indicated top predator was migratory (present <6 months of the year), we considered food-chain length to be the greatest trophic position among non-migratory species, and (iii) we removed multiple instances of the same ecosystem and used only the first instance listed.

Food web data for marine bounded systems were from the database of carefully selected ecosystem network models assembled by Ward et al.[53] (Supplementary Data 2; see Appendix S1 in ref. 53 for network model selection criteria). Marine systems were considered bounded if their connection to adjoining water bodies represented <20% of the total ecosystem perimeter (e.g., embayments, coastal lagoons, and atolls with narrow openings to open ocean systems). For network models, FCL was the maximum trophic position of all groups present in the food web. Within food webs, trophic positions calculated using network models and $\delta^{15}N$ stable isotopes are strongly correlated ($r = 0.69$–$0.99$, with most correlation coefficients >0.85), indicating that estimates of trophic position from network models are reasonably accurate (Appendix S1 in ref. 53). Both stable isotope data and network food web models represented trophic interactions averaged over annual time scales. Marine bounded food webs included birds and marine mammals. Groups present <6 months of the year (e.g., migratory whales, birds and tunas) were removed prior to analysis because transient species are subsidized by production derived from outside focal ecosystems. In cases where some higher order consumers appeared to be excluded from the food web model, the predator group was omitted from biomass pyramid analyses and the system was excluded from analyses of FCL, but remaining data from the ecosystem were included in analyses of C:R and omnivory.

Ecosystem volume was estimated for lakes and marine bounded systems as a hyperbolic sinusoid ($0.43 \times$ area $\times$ maximum depth[20]), or as area $\times$ mean depth when maximum depth was not known. For lakes, where phytoplankton represent the largest contribution to primary productivity and where phosphorus availability is the dominant driver of phytoplankton production[74], ecosystem productivity was estimated using Total Phosphorus concentration (TP; derived from source publications or from a search for mid-Summer data collected in the same year or within a few years of food web sampling). For marine bounded systems, where productivity is derived from phytoplankton, but with important contributions from benthic macrophytes, epiphytic producers, and epibenthic producers, we used the summed annual primary production (t WW km$^{-2}$ yr$^{-1}$) to represent ecosystem productivity.

Lakes were classified as oligo-, meso-, or eutrophic after Carlson[75] (Oligotrophic: TP <10 µg L$^{-1}$, Mesotrophic: TP 10–24 µg L$^{-1}$, Eutrohpic: TP >24 µg L$^{-1}$). Oligo- and mesotrophic lakes were pooled for analyses because they showed similar responses to environmental gradients. Lakes with volume <$10^{-1}$ km$^3$ were classified as small and those with volume >$10^{-1}$ km$^3$ were classified as large.

Productivity and ecosystem volume were not correlated among marine bounded systems ($p = 0.87$, $n = 11$). Although productivity and ecosystem volume were negatively correlated among lake systems ($r = -0.39$, $p = 0.001$, $n = 63$), they were not correlated within categories of low ($p = 0.12$, $n = 40$) and high productivity ($p = 0.51$, $n = 23$) lakes, nor among large lakes ($p = 0.54$, $n = 30$) — the scales of relevance for evaluating the Energy Flux theory. Productivity and ecosystem volume were negatively correlated among small lakes ($r = -0.37$, $p = 0.033$, $n = 33$); however, we do not document any significant drivers of FCL among small lakes; as such, this correlation does not impact our empirical evaluation of the Energy Flux theory.

**Evaluating support for the Energy Flux mechanism**. We evaluated predictions for biomass pyramid shape and omnivory (Fig. 1) in marine bounded systems (we lacked biomass pyramid and diet data for lakes), and predictions for FCL (Figs.1, 2c, e) in marine bounded and lake systems using linear models.

Predictions for biomass pyramid shape were evaluated in marine bounded systems using log ratios of C:R and P:C biomass. Our basal resource group consisted of all autotrophs (detrital biomass data were often unavailable, however, autotroph and detritus biomass are generally positively related, which would enhance any observed patterns in C:R ratios). Following arguments in Ward et al.[53] (their Appendix S2), our consumer group consisted of all consumers with trophic position 2–2.5 (i.e., primary consumers) and our predator group consisted of all consumers with trophic position >2.5 (top predators and mesopredators were

grouped because both groups consumed primary consumers). Because fish species composition was not constant across our productivity and ecosystem size gradients, it was not possible to evaluate the diet of a single fish species across these gradients. Instead, predictions for omnivory were evaluated using an aggregate group of fish with common length 35–40 cm (lengths were taken from fishbase.org; common length is the greatest value of the population length frequency distribution). For each food web, the diets of all fish within this length category were tabulated and omnivory was calculated as the weighted mean (according to consumer biomass) of the fraction of consumer diets comprised of basal resources (autotrophs and detritus).

**Data availability**. All data are available in Supplementary Data 1, 2.

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

## Acknowledgements

We thank T. Tunney and E. Harvey for helpful comments and discussion. C.L.W. was supported by an Ontario Graduate Scholarship in Science and Technology. K.S.M. was supported by NSERC Discovery and Accelerator grants and an NSERC Canada Research Chair in Biodiversity.

## Author contributions

C.L.W. and K.S.M. designed the analysis. C.L.W. collected data and performed empirical and isocline analyses. K.S.M. performed theoretical simulations. C.L.W. wrote the first draft of the manuscript and K.S.M. contributed substantially to revisions.

## Additional information

**Competing interests:** The authors declare no competing financial interests.

