## [Peer Review File · Nature Communications]

Reviewers' comments:

Reviewer #1 (Remarks to the Author):

This manuscript raises the possibility that two mechanisms – species richness and energy flux - may underlie variation in FCL. I found the manuscript interesting but I also found it a bit oddly constructed and hard to reconcile with the existing literature.

Specific comments

1. There are other papers that suggest why researchers may be observing different patterns of FCL with size, production, etc. – including a theoretical paper or two. Of the top of my head, the Takimoto et al. paper is particularly relevant and addresses some very similar ideas, but it is not well integrated into the introduction (just raised in the discussion). Also, I believe a number of the previous papers from the second author make some theoretical predictions that would help set the context a bit more fully.
2. Part of the problem is in the structure of the manuscript. The intro is very short then the manuscript move through some sections that continue to provide context for the core question as they layout specific hypotheses, approaches, and models. I had to wander across much of the manuscript before I could fully follow the development of the ideas, predictions and theory.
3. The introduction sets up the idea that species richness is a commonly discussed mechanism and that this manuscript did not find any empirical support for the role of species richness. I have two major problems with this argument. First, there are not many authors that believe specie richness is important. Most have pointed out that species richness is necessary but not sufficient to explain variation in food chain length. That indicates a strong relationship is very unlikely. Second, if there is a link to species richness, then fish species richness – which is used here and in other test of this idea – is a really poor measure of species richness. Fish are a small proportion of the species found in the food webs of lakes. While fish may be important for determining food web structure in many ways, so are many invertebrate species. There is little value in comparing fish species richness to food chain length.
4. I understand the value to pulling data from the Vander Zanden and Fetzer paper, but that paper has some problems. One is they summarized data across many geographic regions with different species pools – each with its own set of relationships. From this analysis, the strong size effects observed in many studies are minimized in the Vander Zanden and Fetzer paper – Simpson's paradox.
5. Line 27, you cite (25,26) papers that predate the emergence of ecosystem size as an important driver of food chain length as addressing threshold effects (in a sentence about ecosystem size). There have been a couple papers that show no relationship between ecosystem size and food chain length (24 is a good example) but the Warfe paper – despite its title – does not really have the ability to test the ecosystem size effect. It is flawed in a number of ways.

Reviewer #2 (Remarks to the Author):

Ward and McCann develop a theory for food chain length that may explain context dependency between two classical drivers of FCL, namely energy availability and ecosystem size, that are variously supported in the recent literature. The energy flux mechanism that is the basis of their theory is demonstrated using L-V food chain models in a system with three trophic levels including omnivory. The authors test expectations from their model using datasets from lakes and bounded marine ecosystems. I applaud the authors for developing a new theoretical framework to address

a highly relevant and challenging topic. However, I felt many of the assumptions lack sufficient empirical support and the model itself may be too simplistic to be relevant for most systems. More broadly speaking, I think it is important to consider that the context-dependency that this model seeks to explain may be just as likely due to logistical issues in robustly testing food chain predictions in real ecosystems (e.g. how to quantify ecosystem size or energy availability?), and also to question whether a single "unifying framework" is a realistic target given the number of different drivers of FCL that have been supported to various degrees (a truly unified framework would likely require more components, e.g. ecosystem type, evolutionary history/assembly of the species pool, size-structure of predators and prey, relative importance of production sources and degree of benthic-pelagic coupling). Even if a single so-called 'unifying framework' is feasible, to be functional it should at the very least be able to avoid pitfalls of the previous hypotheses, such as how to objectively quantify energy availability for systems when we know that cross-boundary fluxes are important drivers of ecosystem dynamics, how to quantify ecosystem size for systems other than lakes and islands (again, even those are not truly 'closed' due to cross-boundary fluxes), and I would hope it could also account for differences among resource types at least at the scale of 'fast' vs 'slow' pathways which the second author has shown in previous work to be important in structuring food webs and dynamics. To restate, embracing the context-dependency of determinants of food chain length is notable and I agree that it is a promising way forward. However, I don't feel that this particular contribution reaches the heights it suggests due to several methodological limitations in how the model is constructed as well as practical limitations in how such a model represents natural ecosystems and how it would ultimately be tested. Specific comments below should further clarify my opinion.

Paragraph on starting on line 17 – several other factors have also been supported, such as differences among ecosystem types (e.g. papers by Vander Zanden and others cited in the ms), size structure of interactions among different trophic pathways (e.g. Hoeninghaus et al. 2008), others

Line 25 "however, individual tests of environmental drivers in natural settings yield contradictory results" – aside from requiring a more complex model, what about the logistic difficulties of robustly testing the aforementioned hypotheses in natural systems? For example, many of the papers cited are from streams, but how do we quantify the size of a stream in terms of relevance for testing between ecosystem size and energy availability? Few if any of the papers cited follow the same methodology for quantifying ecosystem size or productivity, so how much does that contribute to the lack of consensus that is being used as impetus for this model? Even for more clearly bounded systems such as lakes and islands, many of the authors cited also have works demonstrating the importance of cross-boundary fluxes for food web structure (e.g. work by Spiller and others on marine deposition on islands). In my opinion, this problem of ecosystem bounds or scale is a major issue that the field needs to come to grips with if we presume to advance mechanistic understanding of food web structure. That is not a direct critique of this manuscript but while one side of that coin is to get 'better' data to test existing models, the complement is to formulate new mechanistic hypotheses in a way that facilitates empirical tests.

Sentence starting on line 35 – I wrote in the margin "what about when SR and energy flux are correlated". In fairness, the authors mention exactly this later in the manuscript, but the text could be revised to be more direct to the point.

Line 41 – a truly unifying framework would be applicable to many different ecosystem types, why not add some terrestrial or soil food web datasets?

Line 45 "it is often assumed increasing SR drives rising FCL" – I would disagree with this statement, so perhaps it should be supported more directly with examples or more references. Section (i) – What about type of productivity rather than productivity per se? For example, differences in productivity among sources with differing degrees of palatability or fast vs. slow (following the second author's previous work) have been shown to affect food web structure and

energy flow. The authors come back to the palatability issue later in the text, but that seems to be somewhat circular given that we already know it can play a role.

Line 64 "shifting environmental conditions are often associated with changes in SR" – I would argue that the more common and more significant responses are changes in abundances (e.g. dominance and rarity) and species composition (turnover). Some empirical examples would be useful to support your statement.

L-V models: I have three concerns here: First, why use a three level food chain model? This is shorter than most empirically-determined food chain lengths. Even the datasets used to test the model predictions have FCL ranging between 3 and 5. The model would function quite differently if you add a trophic level or two (and depending on how omnivory is specified in the model, e.g. only to the level below?). Second, why only a single food chain? The second author's work on fast vs slow pathways in food webs has been very influential, why abandon that here? Alternatively, using two pathways may allow for incorporation of a shift in biomass of grazing resistant taxa with increasing productivity that many studies have documented. Incorporating two pathways would not overly complicate the model, while providing options for improving realism. Third, why use a linear function for degree of omnivory rather than a Type II response? Again, a type II response would likely be more realistic.

Line 86: the theory remains qualitatively the same regardless of which approach (modification of model parameters) is employed – ok, but does it remain the same if you add a trophic level to the model or use a more realistic functional response?

Paragraph starting on line 89 – some empirical examples would be useful

Line 100 – did part of this sentence get deleted by accident?

Line 112 – effect of declining ecosystem size – this is functionally different from comparing ecosystems of different sizes. While the approach makes sense if I think about shrinking habitat size increasing the intensity of biotic interactions (e.g. during drought conditions in streams), I'm not confident that the premise should hold across a range of different size ecosystems with associated communities filtered by conditions of each particular system. This is an important consideration because the use of aPC to reflect the effect of ecosystem size is an integral part of the next section (e.g. line 144).

Line 115 – this depends on the number of trophic levels (e.g. Oksanen et al 1981) and which is why I am concerned about the utility of a three level food chain model to predict FCL in systems with much longer food chains.

Paragraph starting on line 121 – I would delete the entire paragraph; it does not add significantly to the main objective of the ms

Sentence starting on line 149 – Any empirical examples to support this? Productivity and ecosystem size were tested in multiple studies of lake food webs, with results contradictory to this statement.

Line 173: "Eventually, the energy flux mechanism can begin to play a role..." – the preceding sections would have been more clear if presented in a similar manner (i.e. when each mechanism is important).

Line 183, (iii) – Yes! But how is this incorporated in the author's "unifying framework"?

Line 195: island biogeography theory – do you mean SR? you have not addressed connectivity...

Line 198 – any empirical examples?

Line 207 – is this incorporated in the model?

Line 211: used this dataset to test the SR mechanism – why not present both the SR and energy flux paragraphs together rather than separating by the marine dataset?

Line 212 – Is fish species richness an appropriate surrogate for community SR? I presume that the other taxonomic groups lower in the food web comprise a much greater component of the diversity in the food web.

Line 230: cannot distinguish between SR and energy flux mechanisms – this whole section seems circular

Line 237: “under these conditions our theory predicts ...” – how do you get from a general model shape (i.e. no specific axis values) to defining how these particular systems fit predictions? This seems quite arbitrary.

Line 255 – how so? Line 238 states that FCL should be weakly or unrelated to ecosystem size

Line 291 “although a recent meta-analysis reported strong positive effects of productivity on FCL in what were likely highly productive ecosystems (Everglades wetlands and neotropical rivers) – This is not correct. If you look into the details of the meta-analysis and those studies, you will see that the neotropical rivers are highly oligotrophic due to upstream impoundments.

Line 294 – ok, so then why use those datasets? Line 297 – any support for local saturation in those systems? What about studies of dynamics of FCL over time within a system? Sentence starting on 303 – doesn't this counter your own argument?

Paragraph starting on line 320 – I think you can delete this entire paragraph. Again, if the logic was flawed, why keep coming back to it as a plausible hypothesis?

Line 334 “In conclusion, we provide the first demonstration, ... the effects of ecosystem size and productivity on FCL can be independent of SR in natural settings” – How? All of your findings were hindered by the inability to robustly distinguish the drivers.

Line 338 – I'm not convinced

Line 351 – correct the delta notation in the formula

Line 357 – if you remove migratory species for SR purposes and/or calculating FCL, ideally you would correct for the biomass flux into and out of the system when quantifying productivity

Line 358 – why not average?

line 371 “transient species are subsidized by production from outside the system” – but so are the local food webs. The connectivity of ecosystems in space and time is an important advance in food web research over the last few decades, so why ignore it in your model (if the goal is to provide a unifying framework)?

Line 430 – why 43-40? How does that relate to the distribution in food webs? What is common length?

Figures – why is the hump-shaped prediction for FCL relative to productivity not represented in the empirical findings?

Reviewer #3 (Remarks to the Author):

This paper is a significant contribution that reinvigorates an area of research that has been relatively inactive for several years, but none-the-less remains of primary importance in discovering any underlying rules which may govern the assembly and persistence of biodiverse communities. The manuscript is well written (with some minor deficiencies outlined below) and the modelling approach is elegant and well articulated. The inclusion of omnivory is particularly pleasing, and the profound effect that this has on the model outcomes is an important point of novelty in its own right. It is my belief that this paper is very timely and is likely to generate a further spike of interest in this area, facilitated by a number of emerging large databases of food webs, and the increasing availability of data on primary productivity.

I found the narrative a compelling one and the development of the two alternative hypotheses generally well done. I do have some concerns about whether all alternative theories have been adequately appraised (see below), although the failure to fully assess these other potential covariates does not weaken the paper substantively.

EFFECTS OF DISTURBANCE ON FOOD WEBS. I was not entirely convinced by the assertion that the energy flux theory is a generalisation of disturbance related theories on food-chain length. While instability in food chains is one mechanism whereby disturbance can affect FCL, a second is that predators tend to be relatively rare and may be more vulnerable to local extinctions in disturbed ecosystems.

EFFECTS OF HABITAT HETEROGENEITY. There is abundant theory about the potential role of refugia in determining energy flow, and to some extent heterogeneity effects are likely to be rolled up into the species richness hypothesis. The two systems modelled are both relatively simple structurally, and there has been some evidence that general rules extracted from limnetic/pelagic systems are not always transferrable to more 2D and heterogeneous environments (e.g. Riede et al 2011, *Ecol Letts* 14:169-178).

ROLES OF SYSTEM OPENNESS and SPATIAL FACTORS. I appreciate that the decisions made on excluding systems with species which are migratory or move in and out of the systems is a pragmatic one. However the export of energy from some systems by these means can not be ignored, and it does highlight the potential for spatial factors to influence these outcomes. For example, predators feeding over very large areas may not readily be associated with patch scale productivity or diversity patterns. Similarly, persistence of predators in systems can be a factor of periodic energy imports not detectable by the kind of energy flux approaches used here.

I think that this paper could make an additional contribution by proposing that an important next step would be to understand the temporal and spatial dynamics which may operate in concert with the local scale drivers which are modelled.

Comments to Reviewers:

We are grateful to the reviewers for pushing us to improve this manuscript.

Reviewer #1:

This manuscript raises the possibility that two mechanisms – species richness and energy flux - may underlie variation in FCL. I found the manuscript interesting but I also found it a bit oddly constructed and hard to reconcile with the existing literature.

Specific comments

1. There are other papers that suggest why researchers may be observing different patterns of FCL with size, production, etc. – including a theoretical paper or two. Of the top of my head, the Takimoto et al. paper is particularly relevant and addresses some very similar ideas, but it is not well integrated into the introduction (just raised in the discussion).

We were remiss not to better integrate the Takimoto et al. (2012) paper. While related, we feel our approach is quite distinct from that taken by Takimoto et al. because (i) we argue that several hypothesized drivers of food chain length can act analogously to the omnivory component of IGP (the strength of which is varied by environmental gradients), rather than in tandem with (and in a manner distinct from) IGP, as the Takimoto paper argues, and (ii) we make predictions in an explicitly local context, while the Takimoto paper does so in an explicitly metacommunity context. As such, we feel the manuscript flows best when a description of the Takimoto et al. paper is incorporated into the discussion.

We have added the following to the Introduction:

Line 41-41: "... drivers of FCL may operate simultaneously or interactively (Takimoto et al. 2012) ...

Line 43-46: "Building on work advocating the importance of omnivory to FCL (Post & Takimoto 2007, Takimoto et al. 2012) ... here we argue that understanding factors which underlie the degree of omnivory ... may offer a tractable way forward."

And we have added the following to the Discussion:

Lines 294-298: "A notable, related approach is that of Takimoto et al. (2012), who made predictions for the simultaneous effects on FCL of colonization / extinction dynamics and the strength of Intraguild Predation across gradients of basal productivity, patch density (as a metric of ecosystem size), and disturbance in an explicitly metacommunity context."

Also, I believe a number of the previous papers from the second author make some theoretical predictions that would help set the context a bit more fully.

We have added the following text to the Introduction:

Lines 43-46: “Building on work advocating the importance of omnivory to FCL (Post & Takimoto 2007, Takimoto et al. 2012) and on work in the FCL-ecosystem size literature (McCann et al. 2005, Tunney et al. 2012) here we argue that understanding factors which underlie the degree of omnivory, a widespread (Thompson et al. 2007) and stabilizing (McCann & Hastings 1997, Vandermeer 2006, Gellner & McCann 2012) property of food webs, may offer a tractable way forward.”

2. Part of the problem is in the structure of the manuscript. The intro is very short then the manuscript move through some sections that continue to provide context for the core question as they layout specific hypotheses, approaches, and models. I had to wander across much of the manuscript before I could fully follow the development of the ideas, predictions and theory.

We agree that the Introduction was quite short. We have added a paragraph introducing the Resource Availability and Ecosystem Size hypotheses (lines 24-33), which are both particularly pertinent to this manuscript. That said, we have kept the Introduction brief because we introduce much of the background information in the course of developing our theory in the Theory section, which immediately follows.

We have also added the following text to the Introduction to help direct readers’ expectations for the manuscript.

Lines 43-46: “Building on work advocating the importance of omnivory to FCL (Post & Takimoto 2007, Takimoto et al. 2012) and on work in the FCL-ecosystem size literature (McCann et al. 2005, Tunney et al. 2012) here we argue that understanding factors which underlie the degree of omnivory ... may offer a tractable way forward.”

3. The introduction sets up the idea that species richness is a commonly discussed mechanism and that this manuscript did not find any empirical support for the role of species richness. I have two major problems with this argument. First, there are not many authors that believe specie richness is important. Most have pointed out that species richness is necessary but not sufficient to explain variation in food chain length. That indicates a strong relationship is very unlikely.

We have incorporated this point into the Supplementary Information 2 section regarding the additive species richness mechanism (which we discuss further below):

Line S2.88 – S2.92: “Theory (Takimoto et al. 2012) and empirical work in natural settings suggest that species richness alone is often necessary yet insufficient to predict FCL, and instead acts in concert with other drivers of FCL (e.g. McHugh et al. 2010, Young et al. 2013). These observations suggest that FCL should not necessarily bear a strong relationship to species richness.”

Although empirical work has failed to identify species richness as a standalone driver of food chain length in natural settings, it remains a valid hypothesis and underlies one mechanism by which environmental gradients may influence food chain length. As such, we feel it is relevant to retain the verbal theory section for the species richness mechanism, as well as the conceptual summary of where species richness should be important to food chain length across environmental gradients (Theory section (v) and Fig. 3).

Second, if there is a link to species richness, then fish species richness – which is used here and in other test of this idea – is a really poor measure of species richness. Fish are a small proportion of the species found in the food webs of lakes. While fish may be important for determining food web structure in many ways, so are many invertebrate species. There is little value in comparing fish species richness to food chain length.

We appreciate this comment. Species richness can influence food chain length via the addition / removal of top predators or insertion of consumers at intermediate trophic levels. We agree that it is questionable whether the species insertion component of the Species Richness hypothesis for FCL is sufficiently addressed by fish species richness alone, in the absence of invertebrate richness data for lower trophic level invertebrates. For this reason, **we have removed the species richness analysis from the main text.**

However, because top predators in the lakes we use are invariably fish, we feel that fish species richness remains a suitable metric to address the additive component of the Species Richness hypothesis. We believe it would be of interest to readers to assess whether our lake ecosystems show evidence of this mechanism dominating FCL patterns. As such, we have placed the species richness analysis in a Supplementary Information section, acknowledging that we work across different regional species pools and assume, like all published tests of the Species Richness hypothesis for FCL, local saturation of species richness. Although we feel this analysis adds to the manuscript, we are happy to remove it entirely if the reviewers have strong reservations.

4. I understand the value to pulling data from the Vander Zanden and Fetzer paper, but that paper has some problems. One is they summarized data across many geographic regions with different species pools – each with its own set of relationships. From this analysis, the strong size effects observed in many studies are minimized in the Vander Zanden and Fetzer paper – Simpson's paradox.

We have added the following text to the discussion:

Lines 338-352: “An additional caveat is that we use data from ecosystems located in different geographic regions with different regional species pools – each, ostensibly, with a unique suite of relationships between FCL and environmental drivers owing to differences in species identity and regional environmental conditions. We assume that the overall direction and gross shape of relationships between ecosystem-level emergent

properties and environmental drivers will be conserved at the between-ecosystem level, albeit with greater variance – for instance, non-linearities may arise at different environmental thresholds and relationship strengths and elevations may differ owing to between-species physiological, behavioural, and ecological differences in responses to environmental change. This increased variance would, presumably, render it more difficult to detect the relationships we predict. As such, it is notable that our predictions for the Energy Flux theory and the attendant context-dependent nature of FCL [including the expected effect of ecosystem size in oligo/mesotrophic systems] are realized in nearly all of our empirical tests in marine and lake ecosystems (Figs. 4 and 5) in spite of this variation. For the case where we failed to find the expected relationship between FCL and ecosystem size in small ecosystems (Fig. 5d), we speculate that this greater variance at the between-region level may have influenced this result.”

5. Line 27, you cite (25,26) papers that predate the emergence of ecosystem size as an important driver of food chain length as addressing threshold effects (in a sentence about ecosystem size). There have been a couple papers that show no relationship between ecosystem size and food chain length (24 is a good example) but the Warfe paper – despite its title – does not really have the ability to test the ecosystem size effect. It is flawed in a number of ways.

We have removed the reference to Warfe et al. from the text describing prior work on ecosystem size (line 37).

Reviewer #2:

Ward and McCann develop a theory for food chain length that may explain context dependency between two classical drivers of FCL, namely energy availability and ecosystem size, that are variously supported in the recent literature. The energy flux mechanism that is the basis of their theory is demonstrated using L-V food chain models in a system with three trophic levels including omnivory. The authors test expectations from their model using datasets from lakes and bounded marine ecosystems. I applaud the authors for developing a new theoretical framework to address a highly relevant and challenging topic. However, I felt many of the assumptions lack sufficient empirical support and the model itself may be too simplistic to be relevant for most systems. More broadly speaking, I think it is important to consider that the context-dependency that this model seeks to explain may be just as likely due to logistical issues in robustly testing food chain predictions in real ecosystems (e.g. how to quantify ecosystem size or energy availability?), and also to question whether a single “unifying framework” is a realistic target given the number of different drivers of FCL that have been supported to various degrees (a truly unified framework would likely require more components, e.g. ecosystem type, evolutionary history/assembly of the species pool, size-structure of predators and prey, relative importance of production sources and degree of benthic-pelagic coupling). Even if a single so-called ‘unifying framework’ is feasible, to be

functional it should at the very least be able to avoid pitfalls of the previous hypotheses, such as how to objectively quantify energy availability for systems when we know that cross-boundary fluxes are important drivers of ecosystem dynamics, how to quantify ecosystem size for systems other than lakes and islands (again, even those are not truly ‘closed’ due to cross-boundary fluxes), and I would hope it could also account for differences among resource types at least at the scale of ‘fast’ vs ‘slow’ pathways which the second author has shown in previous work to be important in structuring food webs and dynamics.

We appreciate this comment and we agree. We have removed the reference to ‘a unifying theory’ from the abstract (the only place where it appeared in the manuscript).

We have also clarified that this theory is most relevant to aquatic ecosystems – we have added ‘aquatic’ to the manuscript title and added the following text to the Introduction:

Lines 54-59: “Although this theory is germane across ecosystem types, its predictions are more likely to be realized in aquatic ecosystems, where strong body size scaling in pelagic food chains (Shurin et al. 2006) and the predominance of ectothermic consumers with low metabolic costs (and thus high trophic conversion efficiency; Yodzis 1984) are likely to promote strong vertical energy flux (deBruyn et al 2007).”

To restate, embracing the context-dependency of determinants of food chain length is notable and I agree that it is a promising way forward. However, I don’t feel that this particular contribution reaches the heights it suggests due to several methodological limitations in how the model is constructed as well as practical limitations in how such a model represents natural ecosystems and how it would ultimately be tested. Specific comments below should further clarify my opinion.

Paragraph on starting on line 17 – several other factors have also been supported, such as differences among ecosystem types (e.g. papers by Vander Zanden and others cited in the ms), size structure of interactions among different trophic pathways (e.g. Hoeninghaus et al. 2008), others

We have added the hypotheses for body size scaling and ecosystem type to the Introduction, where we mention existing hypotheses for food chain length (line 22).

Line 25 “however, individual tests of environmental drivers in natural settings yield contradictory results” – aside from requiring a more complex model, what about the logistic difficulties of robustly testing the aforementioned hypotheses in natural systems? For example, many of the papers cited are from streams, but how do we quantify the size of a stream in terms of relevance for testing between ecosystem size and energy availability? Few if any of the papers cited follow the same methodology for quantifying ecosystem size or productivity, so how much does that contribute to the lack of consensus that is being used as impetus for this model? Even for more clearly bounded systems such

as lakes and islands, many of the authors cited also have works demonstrating the importance of cross-boundary fluxes for food web structure (e.g. work by Spiller and others on marine deposition on islands). In my opinion, this problem of ecosystem bounds or scale is a major issue that the field needs to come to grips with if we presume to advance mechanistic understanding of food web structure. That is not a direct critique of this manuscript but while one side of that coin is to get ‘better’ data to test existing models, the complement is to formulate new mechanistic hypotheses in a way that facilitates empirical tests.

As the reviewer alludes, we feel this issue is beyond the scope of this manuscript. We feel this point is addressed, in part, by removing our reference to a ‘unifying theory’ from the abstract.

We have also added the following text to the Discussion:

Lines 353-367: “We consider the Energy Flux mechanism – and its implications for context-dependency in environmental drivers – within an explicitly local context. However, the spatial landscape structure in which local communities are inherently embedded can interact with local environmental drivers and influence local ecosystem dynamics – and, by extension, FCL – in several ways. (i) Theory suggests that colonization / extinction processes arising from metacommunity patch dynamics can influence predictions for FCL (Holt 1996, Calcagno et al. 2011, Takimoto et al. 2012). (ii) In ecosystems permitting the passage of highly mobile predators foraging over very large areas (e.g. migratory species in marine ecosystems), predator biomass, diet, and foraging behavior may not reflect local environmental conditions. (iii) Many ecosystems traditionally regarded as ‘closed’ routinely receive resource subsidies across ecosystem boundaries (e.g. Polis et al. 1997, Spiller et al. 2010, Cole et al. 2011). Collectively these points highlight a largely intractable issue with defining ecosystem boundaries – and, by extension, accurately quantifying ecosystem size and resource availability. Incorporating this question into our framework is beyond the scope of this work, but we point the reader towards recent work in the metaecosystem literature, which is beginning to address these ideas (Marleau et al. 2010, Gounand et al. 2014).”

Sentence starting on line 35 – I wrote in the margin “what about when SR and energy flux are correlated”. In fairness, the authors mention exactly this later in the manuscript, but the text could be revised to be more direct to the point.

For the sake of clarity and flow, we feel the manuscript reads best when this sentence remains as is. As we state in response to Reviewer 1’s second comment, we have purposely kept the Introduction brief because we describe mechanics and underlying theories in the following Theory section.

Line 41 – a truly unifying framework would be applicable to many different ecosystem types, why not add some terrestrial or soil food web datasets?

As we discuss above, we have clarified that this theory and manuscript are most relevant to aquatic ecosystems.

We have grouped the following six comments as they arise and are addressed, wholly or in part, by a single underlying point.

In the theory section (lines 61 – 232) our goals are to (i) concisely summarize theory as it currently exists in the literature, and (ii) develop our Energy Flux theory. While we appreciate the reviewer’s comment about adding empirical examples to the text, we feel that doing so would detract from the purpose and flow of this section.

Line 45 “it is often assumed increasing SR drives rising FCL” – I would disagree with this statement, so perhaps it should be supported more directly with examples or more references.

Our goal here is to summarize theory as it currently exists in the literature. However, we have replaced this passage with “It has been *posited* that ...” (line 63).

Section (i) – What about type of productivity rather than productivity per se? For example, differences in productivity among sources with differing degrees of palatability or fast vs. slow (following the second author’s previous work) have been shown to affect food web structure and energy flow. The authors come back to the palatability issue later in the text, but that seems to be somewhat circular given that we already know it can play a role.

Our goal in this paragraph is to introduce food chain length – species richness theory as it currently exists in the literature, so we do not wish to delve into this concept here.

However, we have added the following text to section (iii) of the Theory section:

Line 147-149: “Although declining resource palatability with increasing productivity may counter the effect of increasing K, declines in palatability would have to largely outweigh increases in productivity for the outcome to be muted.”

We have also added the following text to the Discussion:

Line 306-309: “[Greater omnivory] may also be more apparent in particular food web compartments - for the case of parallel pelagic and detritus-based energy channels, increasing productivity may be associated with greater detrital mass (Cebrian & Lartigue 2004), resulting in shorter detritus-based food chains.”

Line 64 “shifting environmental conditions are often associated with changes in SR” – I would argue that the more common and more significant responses are changes in

abundances (e.g. dominance and rarity) and species composition (turnover). Some empirical examples would be useful to support your statement.

We agree with the reviewer's comment about relative abundance and turnover. However our goal in this sentence fragment is to provide a segue way for why the rest of this paragraph and manuscript seek to move beyond existing food chain length – species richness theory, which is rooted in species richness per se rather than changing community composition.

Given that our goal in this section is to concisely summarize existing theory, we do not wish to delve into this concept here.

Paragraph starting on line 89 – some empirical examples would be useful

The goal of this paragraph is to propose a novel theory, on which the rest of the manuscript is based. Later in this manuscript we analyze empirical data, which largely agrees with this novel theory. In the interest of clarity and concision, we feel this section should remain concise and restricted to its original purpose, which is the presentation of theory.

Sentence starting on line 149 – Any empirical examples to support this? Productivity and ecosystem size were tested in multiple studies of lake food webs, with results contradictory to this statement.

This sentence refers strictly to our theory result. We have clarified this point by adding “*Our* energy flux theory predicts ...” at the start of the paragraph (line 165). Additionally, we reiterate our response immediately above with respect to Line 89.

Line 198 – any empirical examples?

In the interest of clarity, we feel this section should remain concise and restricted to its original purpose, which is the presentation of the new theory we propose.

L-V models: I have three concerns here: First, why use a three level food chain model? This is shorter than most empirically-determined food chain lengths. Even the datasets used to test the model predictions have FCL ranging between 3 and 5. The model would function quite differently if you add a trophic level or two (and depending on how omnivory is specified in the model, e.g. only to the level below?). Second, why only a single food chain? The second author's work on fast vs slow pathways in food webs has been very influential, why abandon that here? Alternatively, using two pathways may allow for incorporation of a shift in biomass of grazing resistant taxa with increasing productivity that many studies have documented. Incorporating two pathways would not overly complicate the model, while providing options for improving realism. Third, why

use a linear function for degree of omnivory rather than a Type II response? Again, a type II response would likely be more realistic.

Line 86: the theory remains qualitatively the same regardless of which approach (modification of model parameters) is employed – ok, but does it remain the same if you add a trophic level to the model or use a more realistic functional response?

We appreciate the reviewer's point. We have added a Supplementary Information section in which we demonstrate that the theory results are robust to the variations in model structure proposed by the reviewer (a four trophic level food chain, parallel food chains coupled by a predator, and specification of a Type II functional response).

We have also addressed this in the main text:

Lines 118-125: "These results are robust to variation in food web structure and functional response form (Supplemental Information 1). Theory will generally yield this answer as long as any process drives biomass accumulation in the top predator, which, in turn, will tend to produce cascading top down impacts that generate the conditions for increasing omnivory. As such, the result is very general. Moreover, as we explain below, this result is generally robust to additions of weak to moderate density dependence in C and P (bottom-up forcing), and will persist as long as density dependence is not sufficiently strong to prevent biomass build up (top-heaviness) in the biomass pyramid."

Although further variations in model structure are feasible, we feel that exploring further model variation (e.g. how omnivory is specified, shifts in resource palatability with a parallel channel model, as the reviewer mentions) would warrant a separate manuscript.

Line 100 – did part of this sentence get deleted by accident?

We wonder if there is a typo in the line number referenced by the reviewer, as we cannot identify an incomplete sentence in this section of the manuscript.

Line 112 – effect of declining ecosystem size – this is functionally different from comparing ecosystems of different sizes. While the approach makes sense if I think about shrinking habitat size increasing the intensity of biotic interactions (e.g. during drought conditions in streams), I'm not confident that the premise should hold across a range of different size ecosystems with associated communities filtered by conditions of each particular system. This is an important consideration because the use of aPC to reflect the effect of ecosystem size is an integral part of the next section (e.g. line 144).

We agree with the reviewer that this is an assumption of our theory and so have made this assumption clear in the manuscript. Nonetheless, we draw this assumption from longstanding theoretical results, as we describe in text added to the Theory section:

Line 129-144: "Within ecosystems, reductions in ecosystem size below the foraging scale of top predators can increase the top-down pressure of predators on consumers (McCann

et al. 2005, Van de Koppel et al. 2005). We make the assumption here that this argument can be extended to the between-ecosystem effect of changing ecosystem size, and thus assume that the effect of declining ecosystem size on FCL can be captured in our framework by increasing a_{PC} , the attack rate of predators on consumers. Working from first principles, the theory in McCann et al. 2005 and Van de Koppel et al. 2005 makes mathematical arguments that the more mobile a consumer is that feeds in multiple habitat types (e.g., littoral versus pelagic of aquatic ecosystems), the more its average attack rate increases as these spatially distinct macro-habitats become closer to each other or are simply smaller. Effectively, and ultimately, a mobile consumer “views” such a mixed habitat as well-mixed at some reduced spatial scale, thus increasing its average consumption rate relative to a larger, more complex habitat arrangement. Although theoretical in its origin, recent empirical evidence at the between-ecosystem scale supports these arguments – Tunney et al. (2012), working in a between-ecosystem context, found that lake ecosystem food webs appear to become more top-heavy with decreasing lake size, as such an assumption would predict.”

We have also mentioned this as a caveat in the Discussion:

Line 335-338: “We note several caveats to the work presented here. First, as discussed above, we have assumed that the between-ecosystem effect of changing ecosystem size can be represented by a framework for the within-ecosystem effect of declining ecosystem size proposed by Van de Koppel et al. (2005) and McCann et al. (2005).”

Line 115 – this depends on the number of trophic levels (e.g. Oksanen et al 1981) and which is why I am concerned about the utility of a three level food chain model to predict FCL in systems with much longer food chains.

We have addressed this comment above in our discussion of variation of model structure.

Paragraph starting on line 121 – I would delete the entire paragraph; it does not add significantly to the main objective of the ms

We have removed this paragraph.

Line 173: “Eventually, the energy flux mechanism can begin to play a role...” – the preceding sections would have been more clear if presented in a similar manner (i.e. when each mechanism is important).

We appreciate the reviewer’s suggestion. We have reviewed the text and feel the manuscript is most clear when this text remains as is.

Line 183, (iii) – Yes! But how is this incorporated in the author’s “unifying framework”?

We have removed the reference to a unifying framework and have added the following text to the verbal theory section:

Line 147-149: “Although declining resource palatability with increasing productivity may counter the effect of increasing K , declines in palatability would have to largely outweigh increases in productivity for the outcome to be muted.”

Line 195: island biogeography theory – do you mean SR? you have not addressed connectivity...

We have updated the text as follows:

Line 218-219: “Additionally, maximum FCL is lower in small ecosystems owing to greater a_{PC} (pursuant to Fig. 2) and should also be lower if smaller ecosystems have lower species richness.”

Furthermore, we have added to the discussion that we consider our theory in an explicitly local (rather than metacommunity) context (paragraph beginning line 353).

Line 207 – is this incorporated in the model?

This follows from the mechanism for how ecosystem size influences a_{PC} at the extreme upper end of a gradient in ecosystem size (not shown in the isocline figure), and has been suggested by McCann et al. 2005 and Tunney et al. 2012.

Line 211: used this dataset to test the SR mechanism – why not present both the SR and energy flux paragraphs together rather than separating by the marine dataset?

We have removed the species richness section from the manuscript.

Line 212 – Is fish species richness an appropriate surrogate for community SR? I presume that the other taxonomic groups lower in the food web comprise a much greater component of the diversity in the food web.

We have addressed this comment above in our response to Reviewer 1’s third comment.

Line 230: cannot distinguish between SR and energy flux mechanisms – this whole section seems circular

(N.B. This section has been moved to Supplementary Information 2, Lines S2.61 - S2.82)
We do not agree that this paragraph is circular. In the remainder of the paragraph we outline why there is greater support for the ecosystem size than the species richness

hypothesis in low-productivity lakes. Our statement that “Nevertheless, we cannot robustly distinguish between effects of the Additive Species Richness and Energy Flux mechanisms in low-productivity ecosystems” is a conservative assessment of the conclusions which can be drawn from these data. We also note that this comment only refers to our evaluation of support for our theory in low-productivity ecosystems.

Line 237: “under these conditions our theory predicts ...” – how do you get from a general model shape (i.e. no specific axis values) to defining how these particular systems fit predictions? This seems quite arbitrary.

We advocate strongly that theory can give qualitative predictions and that it can be the direction and shape of a predicted response that is of importance, rather than the specific values where a response occurs. What is relevant here is that we distinguish between ecosystems with lower and higher productivity. We make this distinction where there occurs an order of magnitude increase in productivity; we have updated the manuscript text accordingly:

Line 240-242: “Under these higher-productivity conditions (here we distinguish between low and high productivity where there occurs an order of magnitude difference in primary production) ...”

Line 255 – how so? Line 238 states that FCL should be weakly or unrelated to ecosystem size

The following text was formerly at Line 255: “Regardless of which environmental driver was most important, the overall result for marine bounded ecosystems is that biomass pyramid shape, omnivory, and FCL were driven by the magnitude of vertical energy flux.”

We do not feel these statements are contradictory, and feel that the text in the preceding few lines addresses why:

Line 256-262: “Because productivity and ecosystem size were marginally negatively correlated in high-productivity ecosystems for which FCL estimates were available ($p = 0.066$), FCL cannot be ascribed to a unique driver. However, among these high-productivity ecosystems, Akaike’s Information Criterion indicated greater support for the relationship between FCL and productivity than that with ecosystem size ($\Delta AIC_c = 3.99$). Regardless of which environmental driver was most important, the overall result for marine bounded ecosystems is that biomass pyramid shape, omnivory, and FCL were driven by the magnitude of vertical energy flux.”

Line 291 “although a recent meta-analysis reported strong positive effects of productivity on FCL in what were likely highly productive ecosystems (Everglades wetlands and neotropical rivers) – This is not correct. If you look into the details of the meta-analysis

and those studies, you will see that the neotropical rivers are highly oligotrophic due to upstream impoundments.

This text has been removed from the manuscript.

N.B. The text referenced in the following four comments has been moved to Supplementary Information 2, line S2.92 – S2.107.

Line 294 – ok, so then why use those datasets?

(now line S2.92 – S2.94) We feel the remainder of this paragraph addresses the reviewer’s question and justifies our use of these data while clearly documenting underlying assumptions.

Line 297 – any support for local saturation in those systems?

(now line S2.97) Unfortunately we are unaware of regional species richness estimates for these ecosystems. As such, we (and all other published papers which have evaluated the Species Richness hypothesis for FCL) must assume local saturation.

What about studies of dynamics of FCL over time within a system?

We are unaware of any such published work for these systems.

Sentence starting on 303 – doesn’t this counter your own argument?

(now line S2.103) We disagree that this counters our argument. Our argument is precisely that we must assume that local conditions are more important than the regional species pool, and the second half of this sentence discusses how the lake data we use support this assumption.

Paragraph starting on line 320 – I think you can delete this entire paragraph. Again, if the logic was flawed, why keep coming back to it as a plausible hypothesis?

Despite flaws in the original logic, the dynamic stability hypothesis has remained a central organizing idea and hypothesis in food web ecology and the food chain length literature. Moreover, it now figures prominently in the lotic food chain length literature. Here, we propose an alternative mechanism which may underlie this effect. As such, we feel this section is highly relevant and helps place this theory into greater context within the food chain length literature.

We have added text to this section to highlight the link to FCL in stream ecosystems:

Line 314-316: “the Dynamic Instability hypothesis has nevertheless remained a central organizing idea in the FCL literature and is invoked as a mechanism underlying FCL in streams subject to disturbance (e.g. McHugh et al. 2010, Sabo et al. 2010).”

Line 324-326: “Working in streams, McHugh et al. reported greater omnivory and declining FCL with increasing disturbance, providing evidence supporting this phenomenon.”

Line 334 “In conclusion, we provide the first demonstration, ... the effects of ecosystem size and productivity on FCL can be independent of SR in natural settings” – How? All of your findings were hindered by the inability to robustly distinguish the drivers.

We have removed all sections relating to species richness, including this sentence, from the manuscript main text. We note that the reviewer’s comment about robustly distinguishing between Energy Flux and Species Richness drivers refers only to low-productivity lake ecosystems, and we have addressed this comment above.

Line 338 – I’m not convinced

We believe this comment summarizes several of Reviewer 2’s comments, which we have addressed above.

The goals of this manuscript are to (i) develop theory for the Energy Flux mechanism for food chain length and (ii) then evaluate the level of support for this theory in empirical systems. We believe this manuscript achieves both. We have updated the text in this paragraph to better reflect the manuscript’s goals:

Line 368-376: “In conclusion, we develop theory to show that FCL should arise from an energy flux mechanism in aquatic ecosystems, the context-dependent nature of which should be readily predicted by simple consumer-resource theory. Using data from lake and marine ecosystems, we find that these predictions are largely realized in natural settings: we demonstrate that rising productivity drives declining FCL in large and high-productivity ecosystems (in contrast to predictions of classical hypotheses) and that ecosystem size determines FCL in low-productivity ecosystems, although we failed to observe that ecosystem size drives FCL in small ecosystems. As such, our theory and results may help to reconcile a large literature of seemingly inconsistent results from aquatic ecosystems.”

Line 351 – correct the delta notation in the formula

We have used standard delta notation as per the stable isotope literature (e.g. Vander Zanden & Rasmussen 1999, Post 2002).

Line 357 – if you remove migratory species for SR purposes and/or calculating FCL, ideally you would correct for the biomass flux into and out of the system when quantifying productivity

The reviewer makes a similar comment above with respect to Line 25. As the reviewer alluded there, we feel this issue is beyond the scope of this manuscript. However, this comment refers to lake ecosystems, and we feel it is fair to assume that measured total phosphorus (our estimate of productivity) reflects equilibrium conditions in lakes.

Line 358 – why not average?

We could have calculated an average value or selected only the first study listed; we chose the latter. This involved only one of 63 ecosystems. We are happy to change this if the reviewer feels strongly that we should do so.

line 371 “transient species are subsidized by production from outside the system” – but so are the local food webs. The connectivity of ecosystems in space and time is an important advance in food web research over the last few decades, so why ignore it in your model (if the goal is to provide a unifying framework)?

We have removed our reference to a unifying framework from the abstract. Moreover, as the reviewer has alluded elsewhere, we feel this is beyond the scope of this manuscript.

However, we appreciate this point and we have added the following text to the discussion:

Line 304-306: “This phenomenon may be more apparent seasonally in response to the often strongly temporal nature of cross-ecosystem resource subsidies and autochthonous resource availability (McMeans et al. 2015).”

Line 353-367: “We consider the Energy Flux mechanism – and its implications for context-dependency in environmental drivers – within an explicitly local context. However, the spatial landscape structure in which local communities are inherently embedded can interact with local environmental drivers and influence local ecosystem dynamics – and, by extension, FCL – in several ways. (i) Theory suggests that colonization / extinction processes arising from metacommunity patch dynamics can influence predictions for FCL (Holt 1996, Calcagno et al. 2011, Takimoto et al. 2012). (ii) In ecosystems permitting the passage of highly mobile predators foraging over very large areas (e.g. migratory species in marine ecosystems), predator biomass, diet, and foraging behavior may not reflect local environmental conditions. (iii) Many ecosystems traditionally regarded as ‘closed’ routinely receive resource subsidies across ecosystem boundaries (e.g. Polis et al. 1997, Spiller et al. 2010, Cole et al. 2011). Collectively these points highlight a largely intractable issue with defining ecosystem boundaries – and, by extension, accurately quantifying ecosystem size and resource availability. Incorporating this question into our framework is beyond the scope of this work, but we point the reader towards recent work in the metaecosystem literature, which is beginning to address these ideas (Marleau et al. 2010, Gounand et al. 2014).”

Line 430 – why 43-40? How does that relate to the distribution in food webs? What is common length?

We have added the following text to the manuscript:

Line 450: “... common length is the greatest value of the population length frequency distribution.”

FishBase.org will add this definition to their online Glossary.

We initially chose the 33-40cm length stanza because it corresponded to fish with a movement scale of 100 – 110 km (back-calculated from the relationship between movement scale and body size from Fig. 5a in Rooney et al. 2008 (Ecol. Lett., 11:867-881; doi: 10.1111/j.1461-0248.2008.01193.x). We had grouped fish into 10km bins of movement scale and found that most food webs had fish falling in the 100-110km movement scale.

However, it is not necessary to select fish by movement scale. If we instead group fish into 5cm size bins and select the bin which (i) is present in as many food webs as possible, and (ii) in which most fish have trophic position ≥ 3 (i.e. clearly belonging to an upper trophic level), we again select a similar size bin (35-40cm). We have substituted omnivory estimates for this size bin into our analysis (Figs. 4c, d).

We have not calculated how either of these length stanzas relates to the distribution in food webs because this information would not influence our choice of length stanza; what was important was that the size class was present in most food webs and represented fish capable of belonging to an upper trophic level.

Figures – why is the hump-shaped prediction for FCL relative to productivity not represented in the empirical findings?

As we state in the caption of Fig. 3, the ascending portion of the proposed unimodal relationship should arise among ecosystems where the Species Richness mechanism is the dominant driver of food chain length, and the descending portion should arise among ecosystems where the Energy Flux mechanism is dominant.

We believe this comment refers to Fig. 5 (empirical data from lakes). These data meet three of four predictions of the Energy Flux theory, suggesting that the Energy Flux mechanism for food chain length was dominant among these ecosystems.

Reviewer #3:

This paper is a significant contribution that reinvigorates an area of research that has been relatively inactive for several years, but none-the-less remains of primary importance in discovering any underlying rules which may govern the assembly and persistence of biodiverse communities. The manuscript is well written (with some minor deficiencies outlined below) and the modelling approach is elegant and well articulated. The inclusion of omnivory is particularly pleasing, and the profound effect that this has on the model outcomes is an important point of novelty in its own right. It is my belief that this paper is very timely and is likely to generate a further spike of interest in this area, facilitated by a number of emerging large databases of food webs, and the increasing availability of data on primary productivity.

I found the narrative a compelling one and the development of the two alternative

hypotheses generally well done. I do have some concerns about whether all alternative theories have been adequately appraised (see below), although the failure to fully assess these other potential covariates does not weaken the paper substantively.

We appreciate the reviewer's point and believe that the spirit of this comment and the following comments is largely addressed by removing the reference to 'a unifying theory' from the abstract. We have also incorporated the following points into the discussion:

EFFECTS OF DISTURBANCE ON FOOD WEBS. I was not entirely convinced by the assertion that the energy flux theory is a generalisation of disturbance related theories on food-chain length. While instability in food chains is one mechanism whereby disturbance can affect FCL, a second is that predators tend to be relatively rare and may be more vulnerable to local extinctions in disturbed ecosystems.

We agree and we have added the following text to the Discussion:

Line 326-328: "An additional mechanism which may contribute to the dynamic instability of long food chains independent of Energy Flux theory is that rarity may render predators more prone to local extinction in disturbed ecosystems."

We have removed the text in the Theory section which stated "The above energy-driven FCL theory is a more subtle version of [the dynamic instability] hypothesis."

Additionally, we have been careful to avoid stating that Energy Flux theory is a generalization of disturbance / dynamic stability theory for FCL, and instead state that the former speaks to the latter.

EFFECTS OF HABITAT HETEROGENEITY. There is abundant theory about the potential role of refugia in determining energy flow, and to some extent heterogeneity effects are likely to be rolled up into the species richness hypothesis. The two systems modelled are both relatively simple structurally, and there has been some evidence that general rules extracted from limnetic/pelagic systems are not always transferrable to more 2D and heterogeneous environments (e.g. Riede et al 2011, *Ecol Letts* 14:169-178).

We have clarified that this theory is most relevant to aquatic ecosystems and systems with low habitat heterogeneity – we have added 'aquatic' to the manuscript title and added the following text to the Introduction:

Lines 54-59: "Although this theory is germane across ecosystem types, its predictions are more likely to be realized in aquatic ecosystems, where strong body size scaling in pelagic food chains (Shurin et al. 2006) and the predominance of ectothermic consumers with low metabolic costs (and thus high trophic conversion efficiency; Yodzis 1984) are likely to promote strong vertical energy flux (deBruyn et al 2007). It is also most likely to be realized in ecosystems with low habitat heterogeneity, where prey refugia do not restrict vertical energy flux."

ROLES OF SYSTEM OPENNESS and SPATIAL FACTORS. I appreciate that the decisions made on excluding systems with species which are migratory or move in and out of the systems is a pragmatic one. However the export of energy from some systems by these means can not be ignored, and it does highlight the potential for spatial factors to influence these outcomes. For example, predators feeding over very large areas may not readily be associated with patch scale productivity or diversity patterns. Similarly, persistence of predators in systems can be a factor of periodic energy imports not detectable by the kind of energy flux approaches used here.

I think that this paper could make an additional contribution by proposing that an important next step would be to understand the temporal and spatial dynamics which may operate in concert with the local scale drivers which are modelled.

We appreciate this point. We have added the following text to the discussion:

Line 304-306: “This phenomenon may be more apparent seasonally in response to the often strongly temporal nature of cross-ecosystem resource subsidies and autochthonous resource availability (McMeans et al. 2015).”

Line 353-367: “We consider the Energy Flux mechanism – and its implications for context-dependency in environmental drivers – within an explicitly local context. However, the spatial landscape structure in which local communities are inherently embedded can interact with local environmental drivers and influence local ecosystem dynamics – and, by extension, FCL – in several ways. (i) Theory suggests that colonization / extinction processes arising from metacommunity patch dynamics can influence predictions for FCL (Holt 1996, Calcagno et al. 2011, Takimoto et al. 2012). (ii) In ecosystems permitting the passage of highly mobile predators foraging over very large areas (e.g. migratory species in marine ecosystems), predator biomass, diet, and foraging behavior may not reflect local environmental conditions. (iii) Many ecosystems traditionally regarded as ‘closed’ routinely receive resource subsidies across ecosystem boundaries (e.g. Polis et al. 1997, Spiller et al. 2010, Cole et al. 2011). Collectively these points highlight a largely intractable issue with defining ecosystem boundaries – and, by extension, accurately quantifying ecosystem size and resource availability. Incorporating this question into our framework is beyond the scope of this work, but we point the reader towards recent work in the metaecosystem literature, which is beginning to address these ideas (Marleau et al. 2010, Gounand et al. 2014).”

Reviewers' comments:

Reviewer #1 (Remarks to the Author):

I find this revised manuscript both interesting and still a bit frustrating. Some of the frustration comes from the structure and the way in which the hypotheses are outlined to setup the theory and empirical test. Overall it is improved.

Specific comments

1) I think this manuscript would be much more effective if it were setup with a set of questions that the manuscript were going to address. The header "Two primary mechanisms" suggest species richness and energy are THE two mechanisms and I – and many others – would not agree with that characterization. I strongly suggest a) refocusing the manuscript to ask how energy flux mechanism might unify mechanisms (or something similar) and b) emphasizing more strongly how this work differs from other similar theoretical papers (i.e., focusing on how factors influence FCL through omnivory). This latter point gets lost at times. I suggest this because the manuscript generally does a good job on the back end of integrating the results into the existing literature but setup continues to frustrates me by not being as clear or explicit as possible (or as needed).

2) I struggled with the significant relationship between FCL and productivity reported on page 14 (pointing to Fig 4e). This felt very much like cherry picking data. The relationship is based on 7 of the 9 data points and the threshold (1000 ww/km²/yr) could have been chosen for a number of reasons. The sample size is just too small to make an argument for separating the data to make this argument. This problem is carried through all of the rest of the arguments about productivity being more strongly supported than ecosystems size (but only for the high-productivity systems).

3) I don't find the conclusions that the data for marine systems is consistent with the magnitude of energy flux given that the predictions were not well supported. This is also a problem with the freshwater lake data which does not fully support the predictions of the energy flow models presented in the manuscript – although the authors provide a plausible explanation in the discussion.

4) I was struck by the conversation of ecosystem boundaries and ecosystem size. I know it is en vogue to argue that all ecosystems are open, but there are a handful of papers on this topic that recognize that the openness of a system depends upon the question being asked and that boundaries should be defined by function (typically at steep boundaries of species interactions or energy and material cycling). This makes this apparently intractable issues tractable and prevents rather inflexible arguments about systems always being open or closed.

Picky comments:

5) L19-23: I still find odd both the list of potential drivers of FCL and a number of the citations. The list mixes a number of related and sometimes nested ideas about determinants of food chain length and some of them have little evidence or the evidence is conflated with other processes. I also found odd a number of the citations. For example, the Fretwell and Oksanen papers are linked to the idea that resources determine food chain length, but that is a necessary assumption of their models not so much a prediction (although it is often viewed as such). Likewise, while the Schoener 1989 paper is great, to link it directly to resources – without noting it for ecosystem size – feels odd and may provide readers with a misimpression of the literature. I had a similar reaction to "recent" hypotheses including idea that emerged in the 1970s and 1980s (e.g., Briand and Cohen) being included among ideas and papers that represent hypotheses supported by much more robust datasets (e.g., those using stable isotope approaches). Finally, I was surprised that, arguably, the most important papers on ecosystem size – and how size relates to other core hypotheses – were not included in the list of papers addressed ecosystems size. I read this as the authors restricting the list to theoretical papers because they are talking about "hypotheses" but

many of the papers listed would have had little impact on this topic without the empirical work that was emerging at the same time. This suggests a rather artificial approach in dividing the literature (theory at the end of the first paragraph and test in the second paragraph) that relates to my comment about the intro.

6) I recognize that the authors have added text to the supplemental information, but I remain VERY skeptical of the species richness mechanism as outlined in this manuscript. Its placement in the manuscript (with the response in the supplemental info) makes this appear First, Holt's sequential trophic dependency model and EHH are both stacked specialists models (as trophic levels in EEH) in which, by definition, the only way to increase food chain length is to increase the number of species. Second, there is very limited evidence that energy or resource availability determines FCL in all but a few relatively special circumstances. Where there is evidence – the strength of the relationship is weak. Under these conditions, species richness may be necessary but not sufficient to explain patterns of FCL. The results from IGP models – noted at the end of this section – of course highlights the simplicity of the earlier models and the potential pit falls of the species richness arguments (FCL can increase and decrease by entire trophic positions in without any changes in species richness when addressed with IGP models)

Reviewer #2 (Remarks to the Author):

I have thoroughly read the revised manuscript and the author responses to reviewer feedback. I still find that some aspects of the manuscript overstate the theoretical basis for the present theory (i.e. the importance of SR), conveniently ignore issues with data quality/comparability from empirical datasets, and are overly vague in regards to outcomes of this model and relevance for actual food webs. I disagree that continued citation of the flawed dynamic stability hypothesis somehow makes it valid and a 'central organizing idea' in lotic food chain literature, but that may be a more philosophical discussion that does not need to be further considered here. The above reservations aside, it is clear that the authors considered my feedback and made substantial effort to address many of my concerns by providing greater clarity in the text and supplemental information. Given that similar points were raised by one of the other reviewers, I think it is safe to suggest that the manuscript benefited from those additions. Beyond that, I am not a co-author and I do not expect the authors to re-structure their models or re-write their manuscript to fit my opinions if they disagree. I think that the scientific community should be allowed to consider this contribution in its present state and I hope that the opinions of others in the community are expressed through rigorous testing of this theory and further refinement of the model. I strongly agree that a context-dependent theory for FCL is urgently needed and that the present theory makes a contribution in this regard.

Check the use of delta to make sure it appears different due to the choice of font rather than being the incorrect symbol (it looks like a partial derivative ∂ rather than the appropriate lowercase delta δ).

Reviewer #3 (Remarks to the Author):

I believe that the authors have responded appropriately and thoughtfully to my earlier comments.

Reviewers' comments:

Reviewer #1 (Remarks to the Author):

We are grateful to Reviewer 1 for thoughtful and constructive comments; we feel these have greatly improved the manuscript.

I find this revised manuscript both interesting and still a bit frustrating. Some of the frustration comes from the structure and the way in which the hypotheses are outlined to setup the theory and empirical test. Overall it is improved.

Specific comments

1) I think this manuscript would be much more effective if it were setup with a set of questions that the manuscript were going to address. The header “Two primary mechanisms” suggest species richness and energy are THE two mechanisms and I – and many others – would not agree with that characterization.

We have removed the following text from the Abstract:

Formerly Lines 3-4: “We posit that these hypotheses arise from two underlying drivers of FCL: species richness and the magnitude of vertical energy flux.”

We have also restructured the Introduction and Theory sections to reflect the reviewer’s suggestions:

1) The Theory section now deals only with the Energy Flux mechanism:

- We have changed the heading of the Theory section from “Two Primary Mechanisms underlying Aquatic Food Chain Length” to “An Energy Flux Mechanism for Aquatic Food Chain Length” (line 77).

- We have also removed the subsection titled “i) The Species Richness Mechanism for FCL” (formerly lines 62-77).

2) We have added the following text as background material to the Introduction:

Lines 33-49:

“Early theory for FCL assumed that communities are structured as relatively simple linear food chains of dietary specialists and, by extension, that vertical change in species richness (additions of top predators to trophic chains) is the main mechanism underlying variation in FCL (e.g. refs. 5, 32, 33). Post & Takimoto¹³ later added that species insertions to trophic chains can also elongate FCL. This mechanism (henceforth the Classical Species Richness mechanism; CSRM) underlies, either wholly or in part, hypotheses related to environmental drivers of FCL. The resource availability hypothesis assumes that energy transfers between trophic levels are inherently inefficient and therefore limiting to the persistence of higher-order consumers. Rising energy inputs to basal trophic levels, or factors improving the energetic efficiency of consumers at intermediate trophic levels, should therefore result in the addition of successive top trophic levels^{2-5,7}. Any positive relationship between productivity and FCL should be driven by species richness unless, among low-productivity ecosystems, rising

productivity renders it more beneficial for a predator to feed at intermediate instead of lower trophic levels. The ecosystem size hypothesis is also consistent with the CSR, although the effect of size may be attributable to other mechanisms. Larger ecosystems often harbour greater species richness, increasing the occupancy likelihood of a novel top predator or intermediate trophic-level consumer capable of elongating food chains^{10, 15, 16}.”

I strongly suggest a) refocusing the manuscript to ask how energy flux mechanism might unify mechanisms (or something similar)

We have updated the Abstract to reflect the reviewer’s suggestion:

Lines 3-4: “Here we posit that the magnitude of vertical energy flux in food webs underlies several drivers of FCL.”

We have added the following text to the Introduction:

Lines 50-61:

“Since development of the CSR, theoretical and empirical work has shown that simple linear food chains are not the dominant structures underlying community organization^{34, 35} and, moreover, are not accurate models for community response to changing environmental conditions³⁴. Instead, omnivory and trophic complexity are prevalent³⁶⁻³⁸, food web structure is flexible³⁹⁻⁴³ and community response to environmental change can occur in the absence of changes in vertical species richness (e.g. refs. 25, 27, 44). Such internal change in food web topology and energy flow provides an alternative mechanism by which FCL can vary. Omnivory, in particular, can be important in determining FCL^{11, 13, 14} and can provide a mechanistic link between FCL and environmental conditions^{11, 17, 18, 27}.”

Here we argue that understanding how and under what conditions omnivory responds to environmental gradients may help us understand context-dependency in drivers of FCL and, by extension, contradictory results from tests of FCL drivers in natural settings.”

and b) emphasizing more strongly how this work differs from other similar theoretical papers (i.e., focusing on how factors influence FCL through omnivory). This latter point gets lost at times. I suggest this because the manuscript generally does a good job on the back end of integrating the results into the existing literature but setup continues to frustrates me by not being as clear or explicit as possible (or as needed).

We have expanded on this section in the discussion. We have placed it in the discussion rather than the introduction because the text reflects on specifics of our theory, which we feel are more readily understood after reading the preceding sections of the manuscript. We would be pleased to move this to the Introduction if the Reviewer or Editorial team feels strongly about the matter.

Lines 304-307:

“That FCL is set by a suite of context-dependent drivers was first proposed by Post³¹, who suggested that contingencies in successional history and environmental conditions

can modify food web structure and attendant FCL from expectations derived from simple linear food chain theory.”

Lines 337-351:

“Our approach adds to a literature documenting mechanisms by which omnivory mediates food web structure and FCL. Building on work suggesting that omnivory (in the context of Intraguild Predation) can determine FCL¹³ across gradients of resource availability^{11, 12}, and following a call-to-arms to recognize that drivers of FCL may be context-dependent³¹, we show that our energy flux theory allows us to predict the simultaneous and context-dependent effects of ecosystem size and resource availability on FCL. Working in an explicitly metacommunity context, Takimoto et al.²⁰¹² made predictions for the simultaneous effects on FCL of colonization / extinction dynamics and the strength of Intraguild Predation across separate gradients of basal productivity, patch density (as a metric of ecosystem size), and disturbance. These authors showed that when Intraguild Predation is strong (in our framework, when biomass pyramids are top-heavy), FCL is shorter than when Intraguild Predation is weak (when biomass pyramids are Eltonian). Here we argue that in an explicitly local context, several hypothesized drivers of food chain length can act analogously to the omnivory component of IGP (the strength of which is varied by environmental gradients), rather than in tandem with (and in a manner distinct from) Intraguild Predation.”

We have also added the following text to the Theory Summary section:

Lines 202-207:

“These qualitative predictions concur with those of Post³¹, who suggested that that productivity should limit FCL only at very low levels and that environmental drivers of FCL may be context-dependent. They also concur with predictions of Post & Takimoto¹³, who suggested that FCL may first increase and then decline with increasing resource availability, pursuant to the predictions of Intraguild Predation theory.”

2) I struggled with the significant relationship between FCL and productivity reported on page 14 (pointing to Fig 4e). This felt very much like cherry picking data. The relationship is based on 7 of the 9 data points and the threshold (1000 ww/km2/yr) could have been chosen for a number of reasons. The sample size is just too small to make an argument for separating the data to make this argument. This problem is carried through all of the rest of the arguments about productivity being more strongly supported than ecosystems size (but only for the high-productivity systems).

We agree that we have very few data for marine systems with which to identify patterns. However, we argue against the cherry picking argument, noting that all marine results for productivity (i.e. Figs. 4a and 4c, in addition to Fig. 4e) are, in fact, consistent with predictions for a dome-shaped relationship between FCL and productivity (i.e. dominance of the Species Richness Mechanism at low productivity and dominance of the Energy Flux mechanism at high productivity, as suggested in Fig. 3a), thus supporting an argument for separating the data into low- and high-productivity ecosystems.

We have added the following text to the Results section to reflect this argument:

Lines 254-265:

“We speculate that the result in Fig. 4e (increasing FCL from low to intermediate values of primary production, and decreasing FCL from intermediate to high values) may arise due to a shift from low to high-productivity conditions. As we note above, results for higher-productivity systems (those with primary production > 1000 t WW/km²/yr) matched predictions of our Energy Flux theory. Although only two food webs had lower productivity (primary production < 400 t WW/km²/yr), we note that results for these systems in Figs. 4 a, c, and e are consistent with predictions of the Classical Species Richness mechanism, which we suggest is dominant over ranges of low productivity (left side of Fig. 3a). Over this range of productivity one would expect to observe increasing ratios of C:R and P:C (as we observe in Fig. 4a) reflecting species additions or insertions, and omnivory remaining low or increasing to very low levels (as we observe in Fig. 4c). However, we note that we have only nine food webs in total for this analysis, rendering it difficult to determine patterns.”

We have also added the following text to the Discussion:

Lines 312 – 324:

“A species richness-based positive relationship between productivity and FCL should manifest only at very low levels of productivity, where productivity limits colonization by predators (Fig. 2a, 3, and as previously suggested by Post³¹). Theory and empirical results suggest that communities will behave more like simple linear food chains – a requisite condition for the CSR mechanism for FCL to be important – at low productivity^{11, 51, 53}. ... **Although we evaluated very few marine ecosystems, it is notable that (i) our marine results (Fig. 4e) concur qualitatively with these expectations and (ii) that the threshold below which there may be a positive relationship between productivity and FCL in our marine ecosystems concurs with the upper limit of the range (100 g C / m² / yr, equivalent to 1000 t WW / km² / yr) proposed by Post³¹.**”

3) I don't find the conclusions that the data for marine systems is consistent with the magnitude of energy flux given that the predictions were not well supported.

We believe that our response to point 2 above addresses this argument.

This is also a problem with the freshwater lake data which does not fully support the predictions of the energy flow models presented in the manuscript – although the authors provide a plausible explanation in the discussion.

The reviewer's point in the latter half of this sentence suggests that this is not problematic.

4) I was struck by the conversation of ecosystem boundaries and ecosystem size. I know

it is en vogue to argue that all ecosystems are open, but there are a handful of papers on this topic that recognize that the openness of a system depends upon the question being asked and that boundaries should be defined by function (typically at steep boundaries of species interactions or energy and material cycling). This makes this apparently intractable issues tractable and prevents rather inflexible arguments about systems always being open or closed.

We have added the following text to the section discussion ecosystem boundaries in the Discussion:

Lines 399-401:

“This treatment will be sufficient for ecosystems where ecological interactions and material cycling are largely constrained to the local scale.”

Picky comments:

5) L19-23: I still find odd both the list of potential drivers of FCL and a number of the citations. The list mixes a number of related and sometimes nested ideas about determinants of food chain length and some of them have little evidence or the evidence is conflated with other processes.

Our intent in this section was to list all existing hypotheses for food chain length.

I also found odd a number of the citations. For example, the Fretwell and Oksanen papers are linked to the idea that resources determine food chain length, but that is a necessary assumption of their models not so much a prediction (although it is often viewed as such).

We appreciate this point and we have updated text in the Introduction accordingly:

Lines 39-43:

“The resource availability hypothesis assumes that energy transfers between trophic levels are inherently inefficient and therefore limiting to the persistence of higher-order consumers. Rising energy inputs to basal trophic levels, or factors improving the energetic efficiency of consumers at intermediate trophic levels, should therefore result in the addition of successive top trophic levels^{2-5, 7}.”

Likewise, while the Schoener 1989 paper is great, to link it directly to resources – without noting it for ecosystem size – feels odd and may provide readers with a misimpression of the literature.

We have removed the Schoener reference from the list of citations for the productivity hypothesis (Line 20), and added the productive space hypothesis to the list of existing hypotheses for FCL:

Lines 22-23:

“ ... and productive space (total ecosystem productivity adjusted for ecosystem size)¹⁰.”

I had a similar reaction to “recent” hypotheses including idea that emerged in the 1970s and 1980s (e.g., Briand and Cohen) being included among ideas and papers that represent

hypotheses supported by much more robust datasets (e.g., those using stable isotope approaches).

We have removed the distinction between classical and recent hypotheses from this section:

Lines 19-23:

“Multiple hypotheses propose a diverse array of potential drivers of FCL, including resource availability¹⁻⁵, the dynamic stability of food web configurations⁶, body size and physiological design constraints⁷, body size scaling⁸, ecosystem type^{9, 10}, intraguild predation¹¹⁻¹⁴, ecosystem size¹⁵⁻¹⁸, and productive space (total ecosystem productivity adjusted for ecosystem size)¹⁰.”

Finally, I was surprised that, arguably, the most important papers on ecosystem size – and how size relates to other core hypotheses – were not included in the list of papers addressed ecosystems size. I read this as the authors restricting the list to theoretical papers because they are talking about “hypotheses” but many of the papers listed would have had little impact on this topic without the empirical work that was emerging at the same time. This suggest a rather artificial approach in dividing the literature (theory at the end of the first paragraph and test in the second paragraph) that relates to my comment about the intro.

It was indeed our intention to first list theoretical arguments for hypotheses and then provide supporting empirical evidence. We do so because we use the latter to emphasize that support for hypotheses is varied and contradictory. We do not believe this distinction downplays the importance of empirical work; instead we believe that empirical work identifies shortcomings in theory, which we attempt to address in this manuscript.

6) I recognize that the authors have added text to the supplemental information, but I remain VERY skeptical of the species richness mechanism as outlined in this manuscript. Its placement in the manuscript (with the response in the supplemental info) makes this appear First, Holt’s sequential trophic dependency model and EHH are both stacked specialists models (as trophic levels in EEH) in which, by definition, the only way to increase food chain length is to increase the number of species.

We appreciate this comment. We have removed this section (formerly Lines 62-77; section titled “The Species Richness Mechanism for FCL”) and added the following paragraph to the Introduction. We believe the latter provides an improved treatment of the species richness mechanism for food chain length:

Lines 33-49:

“Early theory for FCL assumed that communities are structured as relatively simple linear food chains of dietary specialists and, by extension, that vertical change in species richness (additions of top predators to trophic chains) is the main mechanism underlying variation in FCL (e.g. refs. 5, 32, 33). Post & Takimoto¹³ later added that species insertions to trophic chains can also elongate FCL. This mechanism (henceforth the

Classical Species Richness mechanism; CSR) underlies, either wholly or in part, hypotheses related to environmental drivers of FCL. The resource availability hypothesis assumes that energy transfers between trophic levels are inherently inefficient and therefore limiting to the persistence of higher-order consumers. Rising energy inputs to basal trophic levels, or factors improving the energetic efficiency of consumers at intermediate trophic levels, should therefore result in the addition of successive top trophic levels^{2-5,7}. Any positive relationship between productivity and FCL should be driven by species richness unless, among low-productivity ecosystems, rising productivity renders it more beneficial for a predator to feed at intermediate instead of lower trophic levels. The ecosystem size hypothesis is also consistent with the CSR, although the effect of size may be attributable to other mechanisms. Larger ecosystems often harbour greater species richness, increasing the occupancy likelihood of a novel top predator or intermediate trophic-level consumer capable of elongating food chains^{10, 15, 16}.

Second, there is very limited evidence that energy or resource availability determines FCL in all but a few relatively special circumstances. Where there is evidence – the strength of the relationship is weak. Under these conditions, species richness may be necessary but not sufficient to explain patterns of FCL. The results from IGP models – noted at the end of this section – of course highlights the simplicity of the earlier models and the potential pit falls of the species richness arguments (FCL can increase and decrease by entire trophic positions in without any changes in species richness when addressed with IGP models)

We have expanded on the following paragraph in the Discussion about the special circumstances in which evidence exists for a species richness-based positive relationship between resource availability and food chain length:

Lines 310-328:

“The lack of consistent results for effects of productivity may be attributed to variation in the range and level of productivity employed by various studies. A species richness-based positive relationship between productivity and FCL should manifest only at very low levels of productivity, where productivity limits colonization by predators (Fig. 2a, 3, and as previously suggested by Post³¹). Theory and empirical results suggest that communities will behave more like simple linear food chains – a requisite condition for the CSR for FCL to be important – at low productivity^{11, 51, 53}. It is notable that empirical evidence for a positive relationship between productivity and FCL attributed to the CSR derives largely from simple experimental systems with low species richness (e.g. refs. 59, 60) and from natural settings with low or limiting productivity (e.g. geologically young and/or oligotrophic lakes^{28, 61}, arctic tundra^{5, 62}). Although we evaluated very few marine ecosystems, it is notable that (i) our marine results (Fig. 4e) concur qualitatively with these expectations and (ii) that the threshold below which there may be a positive relationship between productivity and FCL in our marine ecosystems concurs with the upper limit of the range (100 g C / m² / yr, equivalent to 1000 t WW / km² / yr) proposed by Post³¹. A negative relationship between productivity and FCL has not been documented elsewhere in the literature, likely because the range over which we

observed a negative relationship ($> 1000 \text{ t WW} / \text{km}^2 / \text{yr}$ in marine ecosystems and $> 24 \text{ ug Total Phosphorus} / \text{L}$ in lakes) has rarely been evaluated, and only then in combination with lower ranges of productivity.”

Lines 334-336:

“In light of the arguments noted above it is perhaps not surprising that we did not find a relationship between fish species richness and FCL [referring to Supplementary Information 2], because most lakes we evaluated were likely not in early successional states and/or limited by productivity.”

Reviewer #2 (Remarks to the Author):

I have thoroughly read the revised manuscript and the author responses to reviewer feedback. I still find that some aspects of the manuscript overstate the theoretical basis for the present theory (i.e. the importance of SR), conveniently ignore issues with data quality/comparability from empirical datasets, and are overly vague in regards to outcomes of this model and relevance for actual food webs. I disagree that continued citation of the flawed dynamic stability hypothesis somehow makes it valid and a ‘central organizing idea’ in lotic food chain literature, but that may be a more philosophical discussion that does not need to be further considered here. The above reservations aside, it is clear that the authors considered my feedback and made substantial effort to address many of my concerns by providing greater clarity in the text and supplemental information. Given that similar points were raised by one of the other reviewers, I think it is safe to suggest that the manuscript benefited from those additions. Beyond that, I am not a co-author and I do not expect the authors to re-structure their models or re-write their manuscript to fit my opinions if they disagree. I think that the scientific community should be allowed to consider this contribution in its present state and I hope that the opinions of others in the community are expressed through rigorous testing of this theory and further refinement of the model. I strongly agree that a context-dependent theory for FCL is urgently needed and that the present theory makes a contribution in this regard.

We do appreciate the reviewer’s comments on this and the previous version of the manuscript. As the reviewer suggests, we prefer to leave the manuscript as is and hope that it will generate constructive discussion of the topic in the literature.

Check the use of delta to make sure it appears different due to the choice of font rather than being the incorrect symbol (it looks like a partial derivative ∂ rather than the appropriate lowercase delta δ).

We are grateful to the reviewer for pointing out this error. We have updated the equation accordingly (Line 433).

Reviewer #3 (Remarks to the Author):

I believe that the authors have responded appropriately and thoughtfully to my earlier comments.

We are grateful to the reviewer for their earlier comments.

REVIEWERS' COMMENTS:

Reviewer #1 (Remarks to the Author):

The authors have done a thorough job addressing my comments. I have no further concerns.